# SHAPE DEFENSE

## ABSTRACT

Humans rely heavily on shape information to recognize objects. Conversely, convolutional neural networks (CNNs) are biased more towards texture. This fact is perhaps the main reason why CNNs are susceptible to adversarial examples. Here, we explore how shape bias can be incorporated into CNNs to improve their robustness. Two algorithms are proposed, based on the observation that edges are invariant to moderate imperceptible perturbations. In the first one, a classifier is adversarially trained on images with the edge map as an additional channel. At inference time, the edge map is recomputed and concatenated to the image. In the second algorithm, a conditional GAN is trained to translate the edge maps, from clean and/or perturbed images, into clean images. The inference is done over the generated image corresponding to the input's edge map. A large number of experiments with more than 10 data sets demonstrate the effectiveness of the proposed algorithms against FGSM, $\ell_\infty$ PGD-40, Carlini-Wagner, Boundary, and adaptive attacks. Further, we show that edge information can a) benefit other adversarial training methods, b) be even more effective in conjunction with background subtraction, c) be used to defend against poisoning attacks, and d) make CNNs more robust against natural image corruptions such as motion blur, impulse noise, and JPEG compression, than CNNs trained solely on RGB images. From a broader perspective, our study suggests that CNNs do not adequately account for image structures and operations that are crucial for robustness. The code is available at: `https://github.com/[masked]`.

## 1 INTRODUCTION

Deep neural networks (LeCun et al., 2015) remain the state of the art across many areas and are employed in a wide range of applications. They also provide the leading model of biological neural networks, especially in visual processing (Kriegeskorte, 2015). Despite the unprecedented success, however, they can be easily fooled by adding carefully-crafted imperceptible noise to normal inputs (Szegedy et al., 2014; Goodfellow et al., 2015). This poses serious threats in using them in safety- and security-critical domains. Intensive efforts are ongoing to remedy this problem.

Our primary goal here is to learn *robust models* for visual recognition inspired by two observations. First, object shape remains largely invariant to imperceptible adversarial perturbations (Fig. 1). Shape is a sign of an object and plays a vital role in recognition (Biederman, 1987). We rely heavily on edges and object boundaries, whereas CNNs emphasize more on texture (Geirhos et al., 2018). Second, unlike CNNs, we recognize objects one at a time through attention and background subtraction (e.g., Itti & Koch (2001)). These may explain why adversarial examples are perplexing.

The convolution operation in CNNs is biased towards capturing texture since the number of pixels constituting texture far exceeds the number of pixels that fall on the object boundary. This in turn provides a big opportunity for adversarial image manipulation. Some attempts have been made to emphasize more on edges, for example by utilizing normalization layers (e.g., contrast and divisive normalization (Krizhevsky et al., 2012)). Such attempts, however, have not been fully investigated for adversarial defense. Overall, how shape and texture should be reconciled in CNNs continues to be an open question. Here we propose two solutions that can be easily implemented and integrated in existing defenses. We also investigate possible adaptive attacks against them. Extensive experiments across ten datasets, over which shape and texture have different relative importance, demonstrate the effectiveness of our solutions against strong attacks. Our first method performs adversarial training on edge-augmented inputs. The second method uses a conditional GAN (Isola et al., 2017) to translate edge maps to clean images, essentially finding a perturbation-invariant transformation.

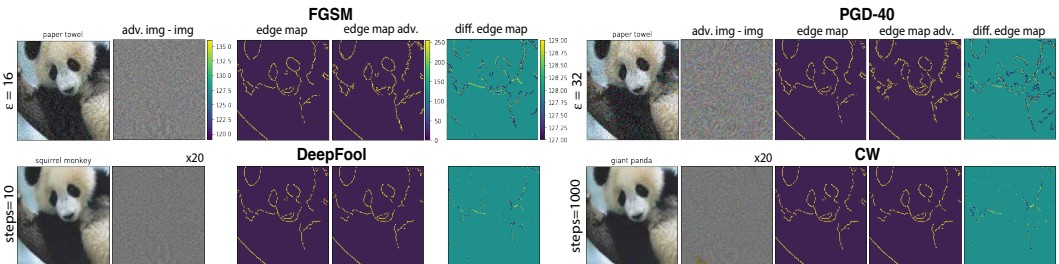

Figure 1: Adversarial attacks against ResNet152 over the giant panda image using FGSM (Goodfellow et al., 2015), PGD-40 (Madry et al., 2017) ($\alpha$=8/255), DeepFool (Moosavi-Dezfooli et al., 2016) and Carlini-Wagner (Carlini & Wagner, 2017) attacks. The second columns in panels show the difference ($\mathcal{L}_2$) between the original image (not shown) and the adversarial one (values shifted by 128 and clamped). The edge map (using Canny edge detector) remains almost intact at small perturbations. Notice that edges are better preserved for the PGD-40. See Appx. A for a more detailed version of this figure, and also the same using the Sobel method.

There is no need for adversarial training (and hence less computation) in this method. Further, and perhaps less surprising, we find that incorporating edges also makes CNNs more robust to natural images corruptions and backdoor attacks. The versatility and effectiveness of these approaches, without significant parameter tuning, is very promising. Ultimately, our study shows that shape is the key to build robust models and opens a new direction for future research in adversarial robustness.

## 2 RELATED WORK

Here, we provide a brief overview of the closely related research with an emphasis on adversarial defenses. For detailed comments on this topic, please refer to Akhtar & Mian (2018).

**Adversarial attacks.** The goal of the adversary is to craft an adversarial input $\tilde{x} \in \mathbb{R}^d$ by adding an imperceptible perturbation $\epsilon$ to the (legitimate) input $x \in \mathbb{R}^d$ (here in the range [0,1]), i.e., $\tilde{x} = x + \epsilon$. Here, we consider two attacks based on the $\ell_\infty$-norm of $\epsilon$, the Fast Gradient Sign Method (FGSM) (Goodfellow et al., 2015), as well as the Projected Gradient Descent (PGD) method (Madry et al., 2017). Both white-box and black-box attacks in the untargeted condition are considered. Deep models are also susceptible to image transformations other than adversarial attacks (e.g., noise, blur), as is shown in Hendrycks & Dietterich (2019) and Azulay & Weiss (2018).

**Adversarial defenses.** Recently, there has been a surge of methods to mitigate the threat from adversarial attacks either by making models robust to perturbations or by detecting and rejecting malicious inputs. A popular defense is **adversarial training** in which a network is trained on adversarial examples (Szegedy et al., 2014; Goodfellow et al., 2015). In particular, adversarial training with a PGD adversary remains empirically robust to this day (Athalye et al., 2018). Drawbacks of adversarial training include impacting clean performance, being computationally expensive, and overfitting to the attacks it is trained on. Some defenses, such as Feature Squeezing (Xu et al., 2017), Feature Denoising (Xie et al., 2019), PixelDefend (Song et al., 2017), JPEG Compression (Dziugaite et al., 2016) and Input Transformation (Guo et al., 2017), attempt to **purify the maliciously perturbed images** by transforming them back towards the distribution seen during training. MagNet (Meng & Chen, 2017) trains a reformer network (one or multiple auto-encoders) to move the adversarial image closer to the manifold of legitimate images. Likewise, Defense-GAN (Samangouei et al., 2018) uses GANs (Goodfellow et al., 2014) to project samples onto the manifold of the generator before classifying them. A similar approach based on Variational AutoEncoders (VAE) is proposed in Li & Ji (2019). Unlike these works which are based on texture (and hence are fragile (Athalye et al., 2018)), our GAN-based defense is built upon edge maps. Some defenses are inspired by biology (e.g., Dapello et al. (2020), Li et al. (2019), Strisciuglio et al. (2020), Reddy et al. (2020)).

**Shape vs. texture.** Geirhos et al. (2018) discovered that CNNs routinely latch on to the object texture, whereas humans pay more attention to shape. When presented with stimuli with conflicting cues (e.g., a cat shape with elephant skin texture; Appx. A), human subjects correctly labeled them based on their shape. In sharp contrast, predictions made by CNNs were mostly based on the texture (See also Hermann & Kornblith (2019)). Similar results are also reported by Baker et al. (2018). Hermann et al. (2020) studied the factors that produce texture bias in CNNs and learned that data augmentation plays a significant role to mitigate texture bias. Xiao et al. (2019), in parallel to our work, have also proposed methods to utilize shape for adversarial defense. They perform classification on the edge map rather than the image itself. This is a baseline method against which we compare our algorithms. Similar to us, they also use GANs to purify the input image.

---

**Algorithm 1** Edge-guided adversarial training (EAT) for $T$ epochs, perturbation budget $\epsilon$, and loss balance ratio $\alpha$, over a dataset of size $M$ for a network $f_\theta$ (performed in minibatches in practice). $\beta \in \{edge, img, imgedge\}$ indicates network type and *redetect_train* means edge redetection during training.

---

**for** $t = 1 \ldots T$ **do**
   **for** $i = 1 \ldots M$ **do**
      *// launch adversarial attack (here FGSM and PGD attacks)*
      $\tilde{x}_i = \text{clip}(x_i + \epsilon \, sign(\nabla_x \ell(f_\theta(x_i), y_i)))$
      **if** $\beta$ == *imgedge* & *redetect_train* **then**
         $\tilde{x}_i = \text{detect\_edge}(\tilde{x}_i)$   *// recompute and replace the edge map*
      **end if**
      $\ell = \alpha \, \ell(f_\theta(x_i), y_i) + (1 - \alpha) \, \ell(f_\theta(\tilde{x}_i), y_i)$   *// here $\alpha = 0.5$*
      $\theta = \theta - \nabla_\theta \ell$   *// update model weights with some optimizer, e.g., Adam*
   **end for**
**end for**

---

---

**Algorithm 2** GAN-based shape defense (GSD)

---

*// Training*
   1. Create a dataset of images $X = \{x_i, y_i\}^{i=1 \cdots N}$ including clean and/or perturbed images
   2. Extract edge maps $(e_i)$ for all images in the dataset
   3. Train a conditional GAN $p_g(x|e)$ to map edge image $e$ to clean image $x$ *// here pix2pix*
   4. Train a classifier $p_c(y|x)$ to map generated image $x$ to class label $y$
*// Inference*
   1. For input image $x$, clean or perturbed, first compute the edge image $e$
   2. Then, compute $p_c(y|x')$ where $x'$ is the generated image corresponding to $e$

---

## 3 PROPOSED METHODS

**Edge-guided Adversarial Training (EAT).** The intuition here is that the edge map retains the structure in the image and helps disambiguate the classification (See Fig. 1). In its simplest form (Fig. 7(A) in Appx. A; Alg. 1), adversarial training is performed over the 2D (Gray+Edge) or 4D (RGB+Edge) input (i.e., number of channels; denoted as Img+Edge). In a slightly more complicated form (Fig. 7(B)), first, for each input (clean or adversarial), the old edge map is replaced with the newly extracted one. The edge map can be computed from the average of only image channels or all available channels (i.e., image plus edge). The latter can sometimes improve the results, since the old edge map (although perturbed; Fig. 10 and Appx. B) still contains unaltered shape structures. Then, adversarial training is performed over the new input. The reason behind adversarial training with redetected edges is to expose the network to possible image structure damage. The loss for training is a weighted combination of loss over clean images and loss over adversarial images. At inference time, first, the edge map is computed and then classification is done over the edge-augmented input. As a baseline model, we also consider first detecting the input's edge map and then feeding it to the model trained on the edges for classification. We refer to this model as Img2Edge.

**GAN-based Shape Defense (GSD).** Here, first, a conditional GAN is trained to map the edge image, from clean or adversarial images, to its corresponding clean image (Alg. 2). Any image translation method (here pix2pix by Isola et al. (2017) using this code[1]) can be employed for this purpose. Next, a CNN is trained over the generated images. At inference time, first, the edge map is computed and then classification is done over the generated image for this edge image. The intuition is that the edge map remains nearly the same over small perturbation budgets (See Appx. A). Notice that conditional GAN can also be trained on perturbed images (similar to Samangouei et al. (2018) and Li & Ji (2019) or edge-augmented perturbed images (similar to above).

## 4 EXPERIMENTS AND RESULTS

### 4.1 DATASETS AND MODELS

Experiments are spread across 10 datasets covering a variety of stimulus types. Sample images from datasets are given in Fig. 2. Models are trained with cross-entropy loss and Adam optimizer (Kingma

---

[1]https://github.com/mrzhu-cool/pix2pix-pytorch

& Ba, 2014) with a batch size of 100, for 20 epochs over MNIST and FashionMNIST, 30 over DogVsCat, and 10 over the remaining. Canny method (Canny, 1986) is used for edge detection over all datasets, except DogBreeds for which Sobel is used. Edge detection parameters are separately adjusted for each dataset. We did not carry out an exhaustive hyperparameter search, since we are interested in additional benefits edges may bring rather than training the best possible models.

The first two datasets include MNIST (LeCun et al., 1998) and FashionMNIST (Xiao et al., 2017). A CNN with 2 convolution, 2 pooling, and 2 fc layers is trained. Each of these datasets contains 60K training images (resolution $28{\times}28$) and 6K test images over 10 classes. The third dataset, DogVsCat[2] contains 18,085 training and 8,204 test images. Images in this dataset are of varying dimensions. They are resized here to $150{\times}150$ pixels to save computation. A CNN with 4 convolution, 4 pooling, and 2 fc layers is trained from scratch.

Over the remaining datasets, we finetune a pre-trained ResNet18 (He et al., 2016), trained over ImageNet (Deng et al., 2009), and normalize images using ImageNet mean and standard deviation.

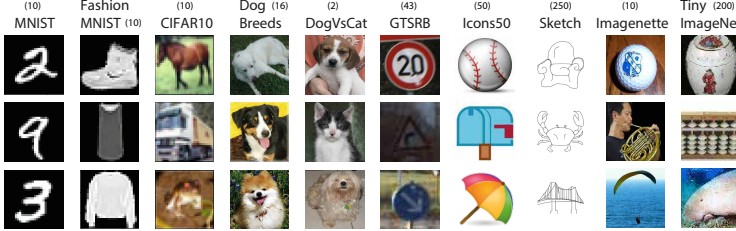

Figure 2: Sample images from the datasets. Numbers in parentheses denote the number of classes.

The fourth dataset, CIFAR-10 (Krizhevsky, 2009), contains 50K training and 10K test images with a resolution of $32{\times}32$ which are resized here to $64{\times}64$ for better edge detection. The fifth dataset is DogBreeds (see footnote). It contains 1,421 training and 356 test images at resolution $224{\times}224$ over 16 classes. The sixth dataset is GTSRB (Stallkamp et al., 2012) and includes 39,209 and 1,2631 training and test images, respectively, over 43 classes (resolution $64{\times}64$ pixels). The seventh dataset, Icons-50, includes 6,975 training and 3,025 test images over 50 classes (Hendrycks & Dietterich, 2019). The original image size is $120{\times}120$ which is resized to $64{\times}64$. The eighth dataset, Sketch, contains 14K training and 6K test images over 250 classes. Images have size $1111{\times}1111$ and are resized to $64{\times}64$ in experiments (Eitz et al., 2012). The ninth and tenth datasets are derived from ImageNet[3]. The Imagenette2-160 dataset has 3,925 training and 9,469 test images (resolution $160{\times}160$) over 10 classes (*tench*, *English springer*, *cassette player*, *chain saw*, *church*, *French horn*, *garbage truck*, *gas pump*, *golf ball*, and *parachute*). The Tiny Imagenet dataset has 100K training images (resolution $64 \times 64$) and 10K validation images (used here as the test set) over 200 classes.

For attacks, we use `https://github.com/Harry24k/adversarial-attacks-pytorch`, except Boundary attack for which we use `https://github.com/bethgelab/foolbox`.

## 4.2 RESULTS

### 4.2.1 EDGE-GUIDED ADVERSARIAL TRAINING

Results over MNIST and CIFAR-10 are shown in Tables 1 and 2, respectively. In these experiments, edge maps are computed only from the gray-level image (in turn computed from the image channels). Please refer to Appx. B for results over the remaining datasets.

Over MNIST and FashionMNIST, robust models trained using edges outperform models trained on gray-level images (the last column). The naturally trained models, however, perform better using gray-level images than edge maps (Orig. model column). Adversarial training with augmented inputs improves the robustness significantly over both datasets, except the FGSM attack on Fashion-MNIST. Over CIFAR-10, incorporating the edges improves the robustness by a large margin against the PGD-40 attack. At $\epsilon = 32/255$, the performance of the robust model over clean and perturbed images is raised from (0.316, 0.056) to (0.776, 0.392). On average, the robust model shows 64% improvement over the RGB model (last column in Table 2). Results when using the Sobel edge detector instead of the Canny does not show a significant difference (Table 7 in Appx. B). Over the TinyImageNet dataset, as in CIFAR-10, classification using edge maps is poor perhaps due to the background clutter. Nevertheless, incorporating edges improves the results. We expect even better

---

[2]www.kaggle.com/c/dogs-vs-cats-redux-kernels-edition & www.kaggle.com/c/dog-breed-identification

[3]https://github.com/fastai/imagenette & https://tiny-imagenet.herokuapp.com

Table 1: Results (Top-1 acc) over MNIST. The best accuracy in each column is highlighted in **bold**. In *italics* are the results of the substitute attack. Epsilon values are over 255. We used the $\ell_\infty$ variants of FGSM and PGD. Img2Edge means applying the Edge model (first row) to the edge map of the image.

| | | Orig. model | | | | Rob. model (8) | | Rob. model (32) | | Rob. model (64) | | Average |
|---|---|---|---|---|---|---|---|---|---|---|---|---|
| | $\epsilon$ | 0/clean | 8 | 32 | 64 | 0/clean | 8 | 0/clean | 32 | 0/clean | 64 | **Rob. models** |
| **FGSM** | Edge | 0.964 | 0.925 | 0.586 | 0.059 | 0.973 | 0.954 | 0.970 | 0.892 | 0.964 | 0.776 | 0.921 |
| | **Img2Edge** | ,, | **0.960** | **0.951** | **0.918** | ,, | **0.971** | ,, | **0.957** | ,, | 0.910 | **0.957** |
| | **Img** | **0.973** | 0.947 | 0.717 | 0.162 | **0.976** | 0.955 | **0.977** | 0.892 | 0.970 | 0.745 | 0.919 |
| | Img+Edge | 0.972 | 0.941 | 0.664 | 0.089 | **0.976** | 0.958 | **0.977** | 0.902 | **0.972** | 0.782 | 0.928 |
| | Redetect | " | 0.950 | 0.803 | 0.356 | " | 0.962 *(0.968)* | " | 0.919 *(0.947)* | " | 0.843 *(0.881)* | 0.941 |
| | **Img + Redetected Edge** | | | | | 0.974 | 0.950 | 0.970 | 0.771 | 0.968 | 0.228 | 0.810 |
| | Redetect | | | | | " | 0.958 *(0.966)* | " | 0.929 *(0.947)* | " | **0.922** *(0.925)* | 0.953 |
| **PGD-40** | Edge | 0.964 | 0.923 | 0.345 | 0.000 | 0.971 | 0.949 | 0.973 | 0.887 | 0.955 | 0.739 | 0.912 |
| | **Img2Edge** | ,, | **0.961** | **0.955** | **0.934** | ,, | **0.970** | ,, | **0.958** | ,, | 0.927 | **0.960** |
| | **Img** | **0.973** | 0.944 | 0.537 | 0.008 | 0.977 | 0.957 | **0.978** | 0.873 | 0.963 | 0.658 | 0.901 |
| | Img+Edge | 0.972 | 0.938 | 0.446 | 0.001 | **0.978** | 0.953 | 0.975 | 0.879 | 0.965 | 0.743 | 0.915 |
| | Redetect | " | 0.950 | 0.741 | 0.116 | " | 0.960 *(0.967)* | " | 0.913 *(0.948)* | " | 0.804 *(0.908)* | 0.932 |
| | **Img + Redetected Edge** | | | | | 0.975 | 0.949 | 0.973 | 0.649 | **0.968** | 0.000 | 0.752 |
| | Redetect | | | | | " | 0.958 *(0.967)* | " | 0.945 *(0.958)* | " | **0.939** *(0.942)* | **0.960** |

Table 2: Results over the CIFAR-10 dataset.

| | | Orig. model | | | Rob. model (8) | | Rob. model (32) | | Average |
|---|---|---|---|---|---|---|---|---|---|
| | $\epsilon$ | 0/clean | 8 | 32 | 0/clean | 8 | 0/clean | 32 | **Rob. models** |
| **FGSM** | Edge | 0.490 | 0.060 | 0.015 | 0.535 | 0.323 | 0.382 | 0.199 | 0.360 |
| | **Img2Edge** | ,, | 0.258 | 0.258 | ,, | 0.270 | ,, | 0.217 | 0.351 |
| | **Img** | **0.887** | 0.359 | 0.246 | **0.869** | 0.668 | **0.855** | 0.553 | 0.736 |
| | **Img + Edge** | 0.860 | 0.366 | 0.169 | 0.846 | 0.611 | 0.815 | 0.442 | 0.679 |
| | Redetect | ,, | **0.399** | **0.281** | ,, | 0.569 *(0.631)* | ,, | 0.417 *(0.546)* | 0.662 |
| | **Img + Redetected Edge** | | | | 0.846 | 0.530 | 0.832 | 0.337 | 0.636 |
| | Redetect | | | | ,, | **0.702** *(0.753)* | ,, | **0.569** *(0.678)* | **0.737** |
| **PGD-40** | Edge | 0.490 | 0.071 | 0.000 | 0.537 | 0.315 | 0.142 | 0.119 | 0.278 |
| | **Img2Edge** | ,, | 0.259 | **0.253** | ,, | 0.274 | ,, | 0.253 | 0.301 |
| | **Img** | **0.887** | 0.018 | 0.000 | 0.807 | 0.450 | 0.316 | 0.056 | 0.407 |
| | **Img + Edge** | 0.860 | 0.019 | 0.000 | 0.788 | 0.429 | 0.176 | 0.119 | 0.378 |
| | Redetect | ,, | **0.306** | 0.093 | ,, | 0.504 *(0.646)* | ,, | 0.150 *(0.170)* | 0.404 |
| | **Img + Redetected Edge** | | | | **0.834** | 0.155 | **0.776** | 0.006 | 0.443 |
| | Redetect | | | | ,, | **0.661** *(0.767)* | ,, | **0.392** *(0.700)* | **0.666** |

results with more accurate edge detection algorithms (e.g., supervised deep edge detectors). Over these 4 datasets, the final model (i.e., adversarial training using image + redetected edge, and edge redetection at inference time) leads to the best accuracy. The improvement over the image is more pronounced at larger perturbations, in particular against the PGD-40 attack (as expected; Fig. 1).

Over the DogVsCat dataset, as in FashionMNIST, the model trained on the edge map is much more robust than the image-only model (Table 8 in Appx. B). Over the DogBreeds dataset, utilizing edges does not improve the results significantly (compared to the image model). The reason could be that texture is more important than shape in this fine-grained recognition task (Table 9 Appx. B). Over GTSRB, Icons-50, and Sketch datasets, *image+edge* model results in higher robustness than the *image-only* model, but leads to relatively less improvement compared to the *edge-only* model. Please see Tables 11, 13, and 15. Over the Imagenette2-160 dataset (Table 17), classification using images does better than edges since the texture is very important on this dataset.

Average results over 10 datasets is presented in Fig. 3 (left panel). Combining shape and texture (full model) leads to a substantial improvement in robustness over the texture alone (5.24% imp. against FGSM and 28.76% imp. against PGD-40). Also, *image+edge* model is slightly more robust than the *image-only* model. Computing the edge map from all image channels improves the results on some datasets (e.g., GTSRB and Sketch) but hurts on some others (e.g., CIFAR-10) as shown in Appx. B. The right two panels in Fig. 3 show a comparison of natural (Orig. model column in tables; solid lines) *vs.* adversarial training. Natural training with *image+edge* and *redetection* at inference time leads to enhanced robustness with little to no harm to standard accuracy. Despite the Edge model only being trained on edges from clean images, the Img2Edge model does better than other naturally-trained models against attacks. The best performance, however, belongs to models trained adversarially. Notice that our results set a new record on adversarial robustness on some of these datasets even without exhaustive parameter search[4].

**Robustness against Carlini-Wagner (CW) and Boundary attacks.** Performance of our method against $l_2$ CW attack on MNIST dataset is shown in Appx. J. To make experiments tractable, we set the number of attack iterations to 10. With even 10 iterations, the original Edge and Img mod-

---

[4]cf. Zhang et al. (2019); the best robust accuracy on CIFAR-10 against PGD attacks is under 60%.

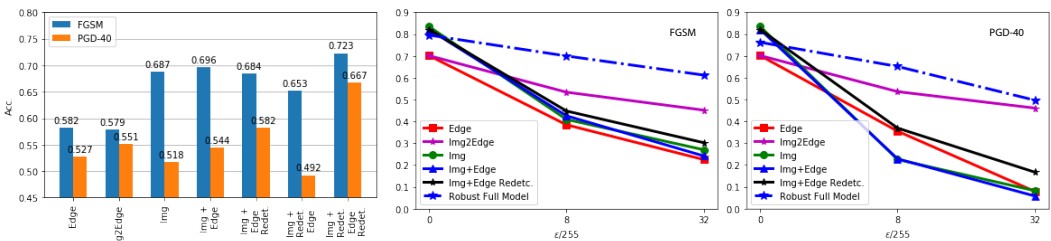

Figure 3: Left) Average results of the EAT defense on all datasets (last cols. in tables). Middle and Right) Comparison of natural (Orig. model column; solid lines) *vs.* adversarial training averaged over all datasets.

els are severely degraded. Img2Edge and Img+(Edge Redetect) models, however, remain robust. Adversarial training with CW attack results in robust models in all cases.

Results against the decision-based Boundary attack (Brendel et al., 2017) are shown in Appx. K over MNIST and Fashion MNIST datasets. Edge, Img, and Img+Edge models perform close to zero over adversarial images. Img+(Edge Redetect) model remains robust since the Canny edge map does not change much after the attack, as is illustrated in Fig. 29.

**Robustness against substitute model attacks.** Following Papernot et al. (2016), we trained substitute models to mimic the robust models (with the same architecture but with RGB channels) using the cross-entropy loss over the logits of the two networks, for 5 epochs. The adversarial examples crafted for the substitute networks were then fed to the robust networks. Results are shown in *italics* in Tables 1, 2, 4 and 5 (performed only against the edge-redetect models). We find that this attack is not able to knock off the robust models. Surprisingly, it even improves the accuracy in some cases. Please refer to Appx. E for more details.

**Robustness against adaptive attacks.** So far we have been using the Canny edge detector which is non-differentiable. What if the adversary builds a differentiable edge detector to approximate the Canny edge detector and then utilizes it to craft adversarial examples? To study this, we run two experiments. In the first one, we build the following pipeline using the HED deep edge detector (Xie & Tu, 2015): Img $\longrightarrow$ HED $\longrightarrow$ Classifier$^{HED}$. A CNN classifier (as above) is trained over the HED edges on the Imagenette2-160 dataset (See Appx. L). Attacking this classifier with FGSM and PGD-5 ($\epsilon = 8/255$) completely fools the network. The original classifier (Img2Edge here) trained on Canny edges, however, is still largely robust to the attacks (i.e., Img$^{adv-HED}$ $\longrightarrow$ Canny $\longrightarrow$ Classifier$^{Canny}$) as shown in Table 29. Notice that the HED edge maps are continuous in the range [0,1], whereas Canny edge maps are binary, which may explain why it is easy to fool the HED classifier (See Fig. 30).

Above, we used an off the shelf deep edge detector trained on natural scenes. As can be seen in Appx. L, its generated edge maps differ significantly from Canny edges. What if the adversary trains a model with the (*input, output*) pair as (*input image, Canny edge map*) to better approximate the Canny edge detector? In experiment two, we investigate this possibility. We build a pipeline consisting of a convolutional autoencoder followed by a CNN on MNIST. Details regarding architecture and training procedure are given in Appx. M. As results in Fig. 33 reveal, FGSM and PGD-40 attacks against the pipeline are very effective. Passing the adversarial images through Canny and then a trained (naturally or adversarially) classifier on Canny edges (i.e., Img2Edge), still leads to high accuracy, which means that transfer was not successful. We attribute this feat to the binary output of Canny. Two important point deserve attention. First, here we used the Img2Edge model, which as shown above, is less robust compared to the full model (i.e., img+edge and redetection). Thus, adaptive attacks may be even less effective against the full model. Second, proposed methods perform better when edge map is less disturbed. For example, as shown in Fig. 33 (bottom), the adaptive attack is less effective against the PGD attack since edges are preserved better.

**Analysis of parameter** $\alpha$. By setting $\alpha = 0$, the network will be exposed only to adversarial examples (Alg. 1), which is computationally more efficient. However, it results in lower accuracy and robustness compared to when $\alpha = 0.5$, which means exposing the network to both clean and adversarial images is important (See Table 19; Appx. D). Nevertheless, here again incorporating edges improves the robustness significantly compared to the image-only case.

**Why is this method working?** The main reason is that the edge map acts as a checksum, and the network learns (through adversarial training) to rely more on the redetected edges when other

channels are misleading (See Table 23). This aligns with prior observations such as shortcut learning in CNNs (Geirhos et al., 2020). Also, our approach resembles adversarial patch or backdoor/trojan attacks where the goal is to fool a classifier by forcing it to rely on irrelevant cues. Conversely, here we use this trick to make a model more robust. Also, the Img2Edge model can purify the input before classifying it. Any adaptive attack against the EAT defense has to alter the edges which most likely will result in perceptible structural damages. See also Figs. 10 & 14 in Appx. A.

### 4.2.2 GAN-BASED SHAPE DEFENSE

We trained the pix2pix model for 10 epochs over MNIST and FashionMNIST, and for 100 epochs over CIFAR-10 and Icons-50 datasets. Sample generated images are shown in Fig. 18 (Appx. F). A CNN (same architecture as before) was trained for 10 epochs to classify the generated images. Results are shown in Fig. 4. The model trained over the images generated by pix2pix (solid lines in the figure) is compared to the model trained over the original clean training set (denoted by the dashed lines). Both models are tested over the clean and perturbed versions of the original test sets of the four datasets. Over MNIST and FashionMNIST datasets, GSD performs on par with the original model on clean test images. It is, however, much more robust than the original model against the attacks. When we trained the pix2pix over the edge maps from the perturbed images, the new CNN models became even more robust (stars in Fig. 4; top panels). We expect even better results with training over edge maps from both intact and perturbed images[5].

Over CIFAR-10 and Icons-50 datasets, generated images are poor. Consequently, GSD underperforms the original model over the original clean images. Over the adversarial inputs, however, GSD wins, especially at high perturbation budgets and against the PGD-40 attack. With better edge detection and image generation methods (e.g., using perceptual loss), even better results are expected.

**Why is this method working?** The main reason is that cGAN learns a function $f$ that is invariant to adversarial perturbations. Since the edge map is not completely invariant to (especially large) perturbations, one has to train the cGAN on the augmented dataset composed of clean and perturbed images. One advantage of this approach is it computational efficiency since there is no need for adversarial training. Any adaptive attack against this defense has to fool the cGAN which is perhaps not feasible since it will be noticed from the generated images (i.e., cGAN will fail to generate decent images).

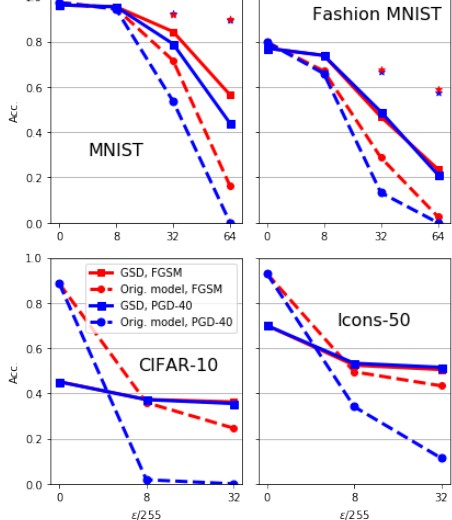

Figure 4: Results of GSD method.

Compared to other adversarial defenses that utilize GANs (e.g., Samangouei et al. (2018); Li & Ji (2019)), our approach relies less on texture. It can be integrated with these defenses.

## 5 FAST & FREE ADVERSARIAL TRAINING WITH SHAPE DEFENSE

Here, we examine whether incorporating shape bias can empower other defenses, in particular, a) fast adversarial training by Wong et al. (2020), dubbed *FastAT*, and free adversarial training by Shafahi et al. (2019), dubbed *FreeAT*. Wong et al. trained robust models using a much weaker and cheaper adversary to lower the cost of adversarial training. They showed that adversarial training with the FGSM adversary is as effective as PGD-based training. The key idea in Shafahi et al. 's work is to simultaneously update both the model parameters and image perturbations in one backward pass, rather than using separate gradient computations at each update step. Please see also Appx. G.

The same CNN architectures as in Wong et al. are employed here. For FastAT, we trained three models over MNIST (for 10 epochs), FashionMNIST (for 3 epochs), and CIFAR-10 (for 10 epochs & *early-stopping*) datasets. For FreeAT, we trained models only over CIFAR-10 for 10 epochs.

Results are shown in Table 3. Using shape-based FastAT and over MNIST, robust accuracy against PGD-50 grows from 95.5% (image-only model) to 98.4% (our full model) at $\epsilon = 0.1$ and from

---

[5]Similarly, the edge map classifier used in the Img2Edge model in the previous section (EAT defense) can be trained on edge maps from both clean and adversarial examples to improve performance.

Table 3: Performance of edge-augmented *FastAT* and *FreeAT* adversarial defenses over clean and perturbed images (See Appx. G for extended algorithms). FastAT is trained with the FGSM adversary ($\epsilon = 0.1$ or $\epsilon = 0.3$) over MNIST and FashionMNIST datasets, and $\epsilon = 8/255$ over CIFAR-10). FreeAT is trained over CIFAR-10 with $\epsilon = 8/255$ and 8 minibatch replays. CIFAR-10 results are averaged over 3 runs (Appx. G). PGD attacks use 10 random restarts. The remaining settings and parameters are as in Wong et al. (2020).

| | MNIST (FastAT) | | | | | | Fashion MNIST (FastAT) | | | | | | CIFAR-10 (FastAT) | | | CIFAR-10 (FreeAT) | | |
|---|---|---|---|---|---|---|---|---|---|---|---|---|---|---|---|---|---|---|
| $\epsilon$ | 0.1 | | 0.3 | | Avg. | | 0.1 | | 0.3 | | Avg. | | 8/255 | | Avg. | 8/255 | | Avg. |
| | 0 | PGD-50 | 0 | PGD-50 | Acc. | | 0 | PGD-50 | 0 | PGD-50 | Acc. | | 0 | PGD-10 | Acc. | 0 | PGD-10 | Acc. |
| Edge | 0.986 | 0.940 | 0.113 | 0.113 | 0.538 | | 0.844 | 0.753 | 0.786 | 0.110 | 0.623 | | 0.582 | 0.386 | 0.484 | 0.679 | **0.678** | **0.678** |
| Img | 0.991 | 0.955 | 0.985 | 0.877 | 0.952 | | 0.835 | 0.696 | 0.641 | 0.000 | 0.543 | | 0.767 | 0.381 | 0.574 | 0.774 | 0.449 | 0.612 |
| Img + Edge | **0.988** | 0.968 | 0.980 | 0.922 | 0.965 | | 0.851 | 0.780 | **0.834** | 0.769 | **0.809** | | **0.874** | 0.386 | 0.630 | **0.782** | 0.442 | 0.612 |
| Redetect | ,, | 0.977 | ,, | 0.966 | 0.978 | | ,, | 0.823 | ,, | 0.778 | 0.822 | | ,, | 0.393 | 0.634 | ,, | 0.448 | 0.615 |
| Img + Red. Edge | 0.986 | 0.087 | **0.986** | 0.000 | 0.515 | | **0.857** | 0.262 | 0.817 | 0.000 | 0.484 | | 0.866 | 0.074 | 0.470 | 0.777 | 0.451 | 0.614 |
| Redetect | ,, | **0.984** | ,, | **0.986** | **0.986** | | ,, | **0.855** | ,, | **0.823** | **0.838** | | ,, | 0.416 | **0.641** | ,, | 0.452 | 0.615 |

87.7% to 98.6% at $\epsilon = 0.3$, which are even higher than what is reported by Wong et al. (97.5% at $\epsilon = 0.1$ and 88.8% at $\epsilon = 0.3$). Over FashionMNIST, the improvement is even more pronounced (from 69.6% to 85.5% at $\epsilon = 0.1$ and from 0% to 82.3% at $\epsilon = 0.3$ ). Over clean images, our full model outperforms other models in most of the cases. Over the CIFAR-10 dataset, the shape-based extension of the defenses results in high accuracy over both clean and perturbed images (using PGD-10 attack), compared to the image-only model. We expect similar improvements with the classic PGD adversarial training. Overall, our analyses in this section suggest that exploiting edges is not specific to the particular way we perform adversarial training (Algorithms 1&2), and be extended to other defense methods (e.g., TRADES algorithm by Zhang et al. (2019)).

## 6 BACKGROUND SUBTRACTION

Background subtraction (a.k.a foreground detection) is an important mechanism by which humans process scenes and recognize objects. It interacts with other mechanisms such as edge and boundary detection. How useful is it for adversarial robustness? In other words, how robust the model will be assuming that the attacker has only access to the foreground object? To find out, we perform an experiment over MNIST and FashionMNIST, for which it is easy to derive the foreground masks. We compare the Img and Edge models (from Section 4.2.1) over the original and noisy (digits placed on white noise background) data, with and without background subtraction and edge detection, against the FGSM attack. Results are shown in Fig. 5(A). First, both models perform poorly over noisy images with the Edge model doing better. Second, post background subtraction, models are much more robust. Third, applying the Edge model to the foreground region leads to almost perfect robustness over MNIST. Even without perfect edge detection, the Edge model does very well over FashionMNIST. This analysis provides an upper bound on the potential benefit from background subtraction on model robustness, assuming that foreground objects can be reliably detected.

## 7 HARNESSING BACKDOOR ATTACKS

Proposed mechanisms can also withstand invisible and visible backdoor attacks (Brown et al., 2017; Liu et al., 2017). Over MNIST, we planted an invisible C-like patch in half of the 8s and relabeled them as 9. We then trained the Img model on this new dataset. The Img model on a test set where all 8s are contaminated (with the patch), classifies almost all of them as 9 (top-left panel in Fig. 5.B). The Edge model, however, correctly classifies them as 8 since edge detection removes the pattern (top-right panel). Thanks to the edge detection, it is also not possible to train the Edge model on the poisoned dataset. A similar experiment on FashionMNIST, using a different patch, shows similar results (bottom panels in Fig. 5.B). In presence of visible patches, the model would not be affected if the correct region is identified (via background subtraction) during training or testing (Appx. I).

## 8 ROBUSTNESS AGAINST NATURAL IMAGE DISTORTIONS

Previous work has shown that ImageNet-trained CNNs generalize poorly over a wide range of image distortions (e.g., Azulay & Weiss (2018); Dodge & Karam (2017)). Our objective in this section is to study whether increasing shape bias improves robustness against common image distortions just as it did over adversarial examples. Following Hendrycks & Dietterich (2019), we systematically test how model accuracies degrade if images are corrupted by 15 different types of distortions including *brightness*, *contrast*, *defocus blur*, *elastic transform*, *fog*, *frost*, *Gaussian noise*, *glass blur*, *impulse noise*, *JPEG compression*, *motion blur*, *pixelatation*, *shot noise*, *snow*, and *zoom blur*, at 5 levels of severity. Fig. 19 (Appx. H) shows sample images along with their distortions.

We test the original models (trained naturally on clean training images) as well as the robust models (trained adversarially using Algorithm 1) over the corrupted versions of test sets on three

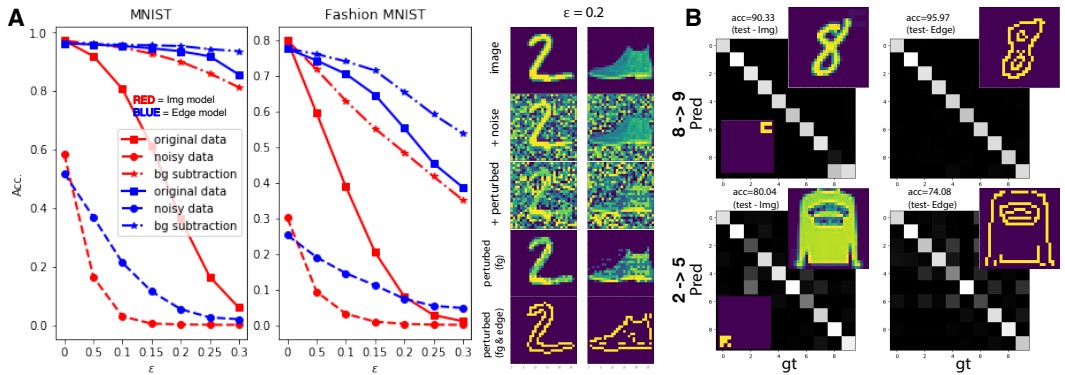

Figure 5: A) Background subtraction together with edge detection improves robustness (here against the FGSM attack). Noisy data is created by overlaying a digit over white noise (noise×(1-mask)+digit). B) Defending backdoor attacks. An almost invisible pattern (with intensity 10/255 of the digit intensity) is added to half of the samples from one class, which are then relabeled as another class. Notice that the Edge model is not confused over the edge maps (right panels) since edge detection removes the pattern. In case of a visible backdoor attack, background subtraction can help discard the irrelevant region. See Appx. I for more details.

datasets. Results are visualized in Fig. 6. See Appx. H for breakdown results on each dataset and distortion. Two conclusions are drawn. First, incorporating edge information in original models (and hence increasing shape bias) improves robustness against common image distortions (solid curves in Fig. 6; RGB+Egde > RGB or Edge). Improvement is more noticeable at larger distortions and over datasets with less background clutter (e.g., Icons-50). This is in alignment with Geirhos et al. (2018) where they showed ResNet-50 trained on the Stylized-ImageNet dataset performs better than the vanilla ResNet-50 on both clean and distorted images.

Second, adversarially-trained models (in particular those trained on Img + Edge) are more robust to image distortions compared to original models. In summary, incorporating edges and adversarial images leads to improved robustness against natural image distortions, *despite models not being trained on any of the distortions during training*. This in turn suggests that the proposed algorithms indeed rely more on shape than texture.

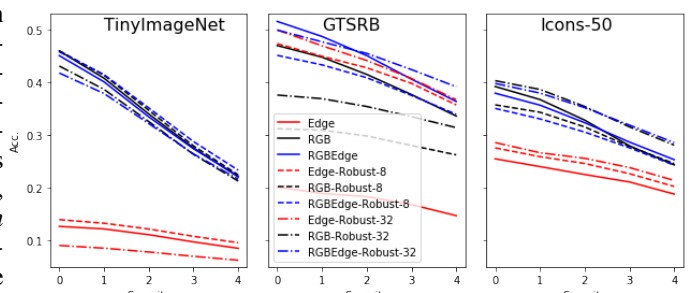

Figure 6: Classification accuracy over naturally distorted images.

## 9 DISCUSSION AND OUTLOOK

Two algorithms are proposed to use shape bias and background subtraction to strengthen CNNs and defend against adversarial attacks and backdoor attacks. To fool these defenses one has to perturb the image such that the new edge map is significantly different from the old one while preserving image shape and geometry, which does not seem to be trivial at low perturbation budgets. Even though we did not perform an exhaustive parameter search (model architecture, epochs, edge detection, cGAN training, etc.), our results are better than or on par with the state of the art in some cases (e.g., over MNIST and CIFAR datasets). The proposed mechanisms are computationally efficient and excel with higher resolution images and low background clutter. They are also more effective against stronger attacks than weaker ones since strong attacks perturb the image less while being more destructive (e.g., PGD *vs.* FGSM; Fig. 1). Shape defense can also be combined with other defenses to produce robust models without a significant slowdown.

Future work should assess shape defense against adversarial attacks such as e.g., gradient-free attacks, decision-based attacks, sparse attacks (e.g., the one pixel attack (Su et al., 2019)), attacks that perturb only the edge pixels, attacks that manipulate the image structure (Xiao et al., 2018), ad-hoc adaptive attacks, , and backdoor (Chen et al., 2017)), as well as other $\ell_p$ norms, and datasets. There might be also other ways to incorporate shape-bias in CNNs, such as 1) augmenting a dataset with edge maps or negative images, 2) overlaying texture from some objects onto some others as in Geirhos et al. (2018), and 3) designing normalization layers (Carandini & Heeger, 2012). Lastly, the interpretation of the shape defense, as in Zhang & Zhu (2019), is another research direction.

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

# A ILLUSTRATION OF SHAPE IMPORTANCE IN ADVERSARIAL ROBUSTNESS

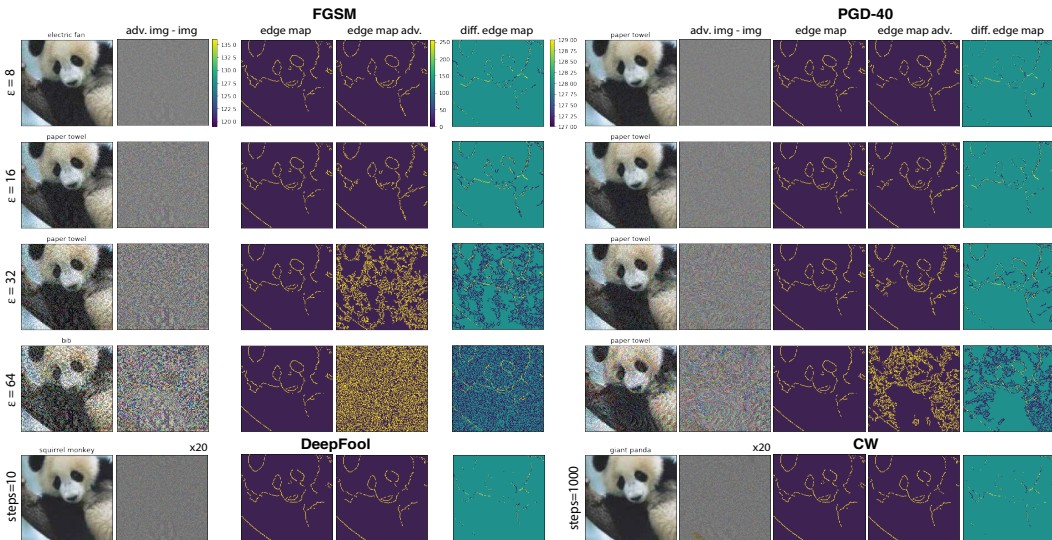

Figure 7: Edge-guided adversarial training (EAT). In its simplest form, adversarial training is performed over the 2D (Gray+Edge) or 4D (RGB+Edge) input (i.e., number of channels; denoted as Img+Edge). In a slightly more complicated form (B), first for each input (clean or adversarial), the old edge map is replaced with the newly extracted one. The edge map can be computed from the average of only image channels or all available channels (i.e., image plus edge).

Figure 8: Adversarial attacks against ResNet152 over the giant panda image using 4 prominent attack types: FGSM (Goodfellow et al., 2015) and PGD-40 (Madry et al., 2017) ($\alpha$=8/255) for different perturbation budgets $\epsilon \in \{8, 16, 32, 64\}$, as well as DeepFool (Moosavi-Dezfooli et al., 2016) and Carlini-Wagner (Carlini & Wagner, 2017). The second column in each panel shows the difference ($\mathcal{L}_2$) between the original image (not shown) and the adversarial one (values shifted by 128 and clamped). For DF and CW, values are magnified 20x and then shifted. The edge map (using the Canny edge detector) remains almost intact at small perturbations. Notice that edges are better preserved for the PGD-40 attack. See Appx. A for results using the Sobel method.

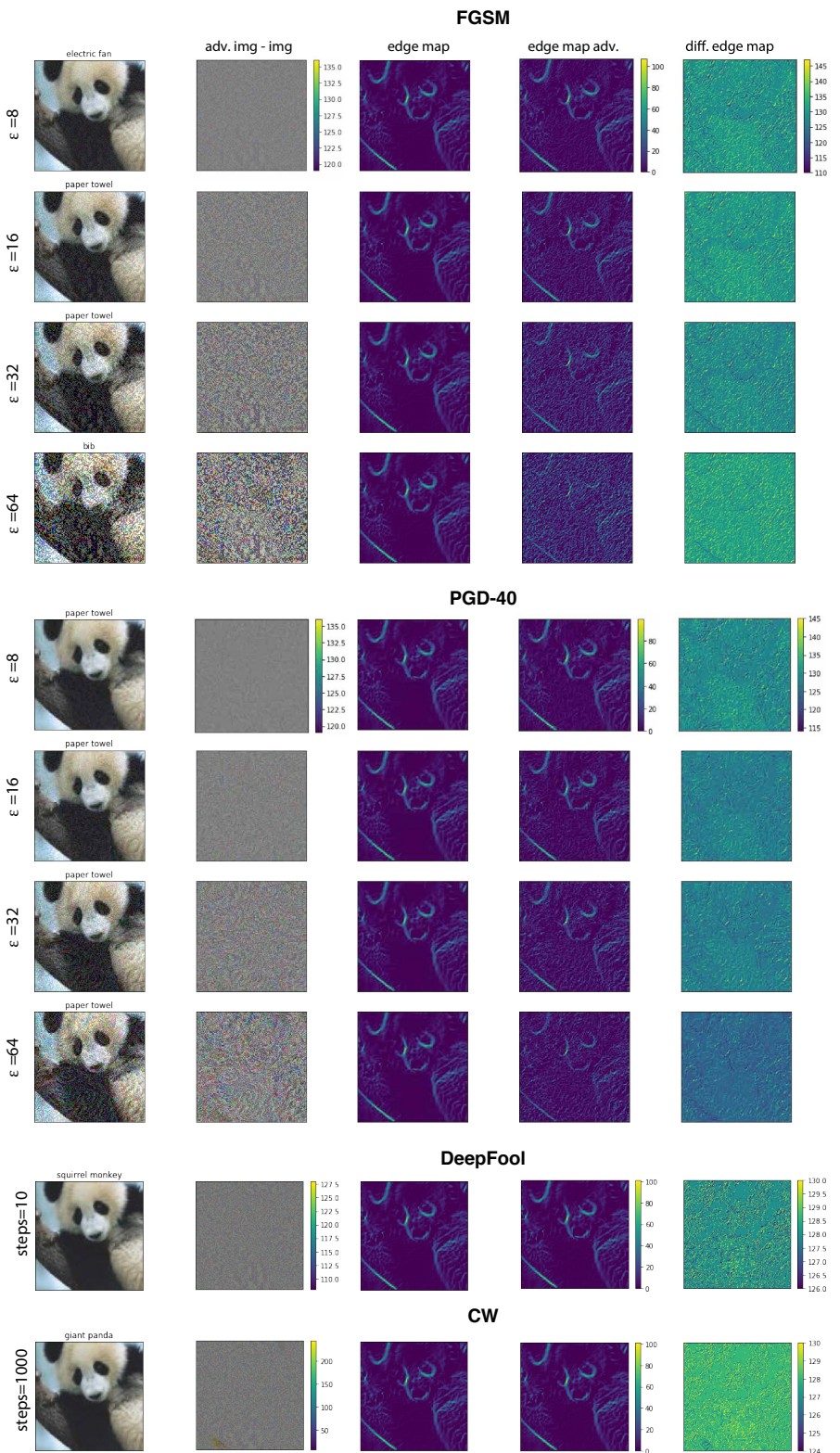

Figure 9: As is in Fig. 1 in the main text but using the Sobel edge detector. As it can be seen edge maps are almost invariant to adversarial perturbation.

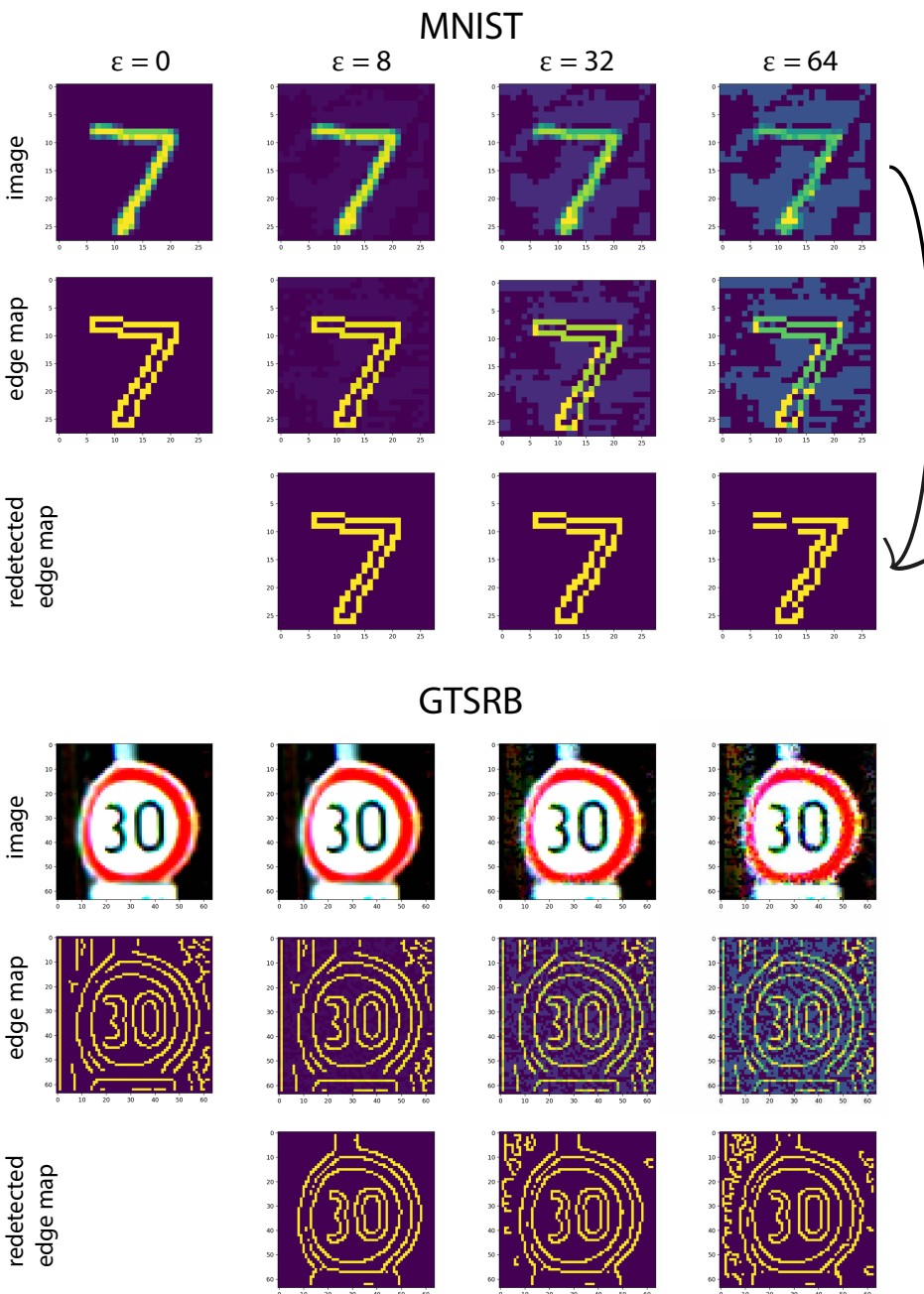

Figure 10: Illustration of adversarial perturbation over the image as well as its edge map. The first row in each panel shows the clean or adversarial image (under the FGSM attack). The second row shows the perturbed edge map (i.e., the edge channel of the the 2D or 4D adversarial input). The third row shows the redetected edge map from the attacked gray or rgb image (i.e., calculated only from the image channels and excluding the edge map itself).

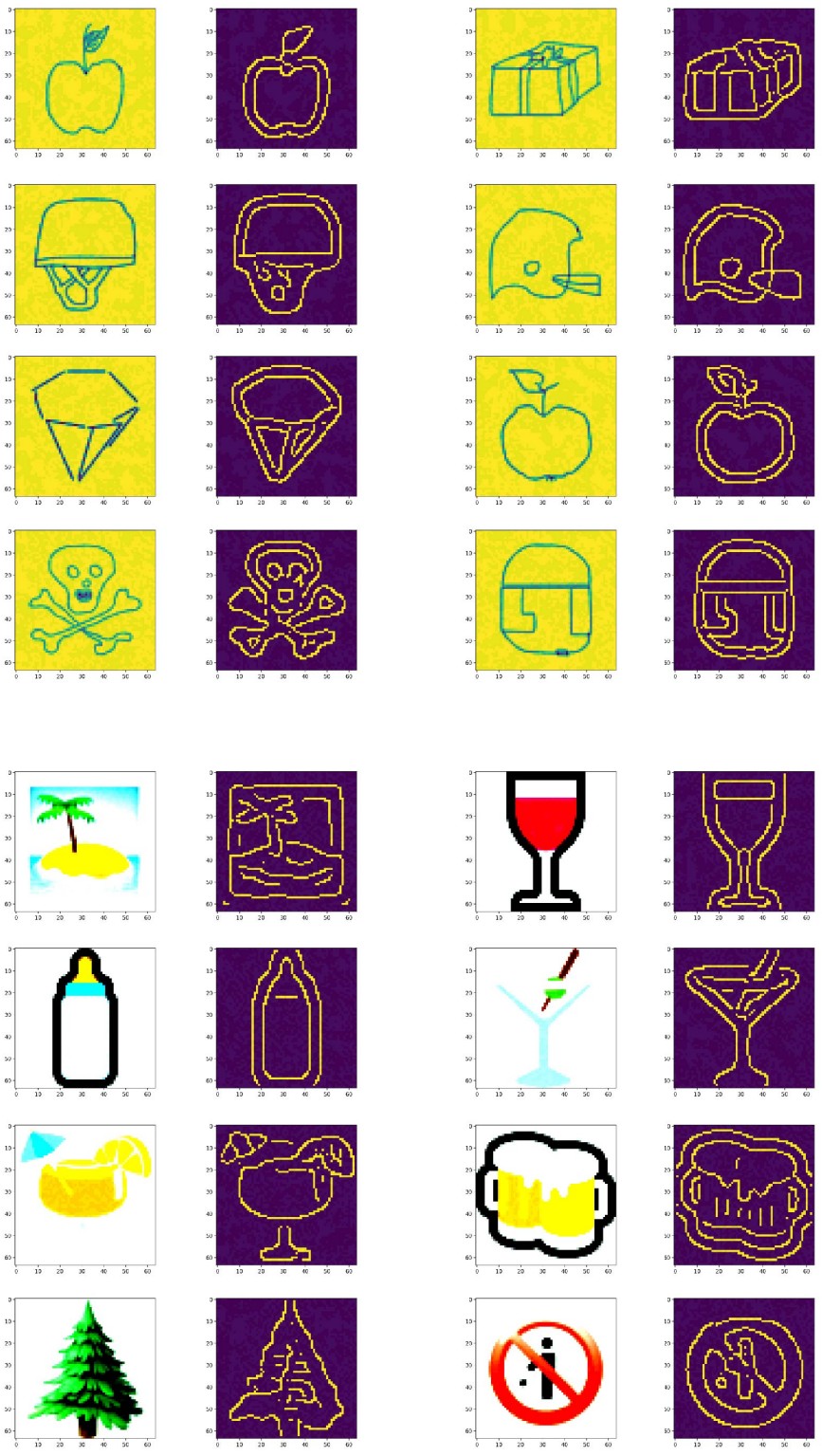

Figure 11: Samples images from Sketch and Icons-50 datasets, perturbed with FGSM $\epsilon = 8/255$, and their corresponding edge maps using Canny edge detection.

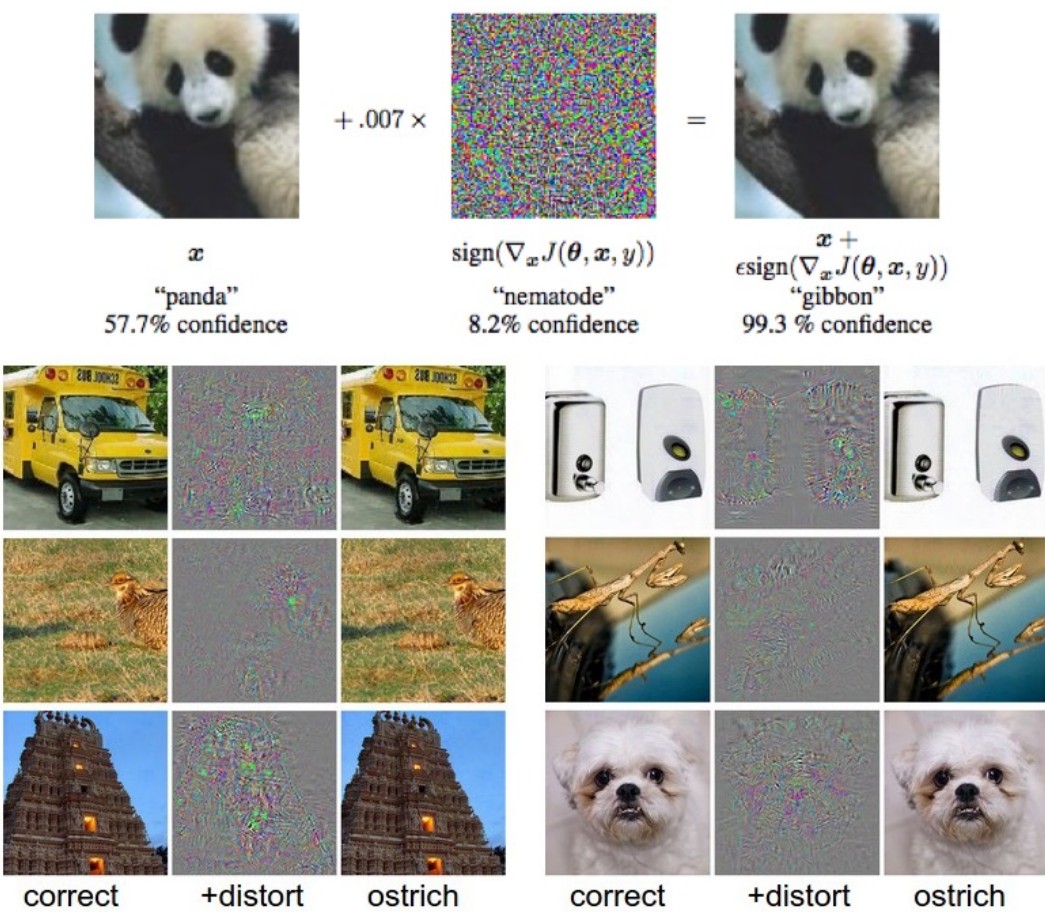

Figure 12: Top) Adversarial example generated for the giant panda image using the FGSM attack (Goodfellow et al., 2015). Bottom) Adversarial examples generated for AlexNet from Szegedy et al. (2014). (Left) is a correctly predicted sample, (center) difference between correct image, and image predicted incorrectly magnified by 10x (values shifted by 128 and clamped), (right) adversarial example (i.e., left image + middle image). Even though the left and right images appear visually the same to humans, the left images are correctly classified by a DNN classifier while the right images are misclassified as "ostrich, Struthio camelus". Notice that in all of these images the overall image structure and edges are preserved.

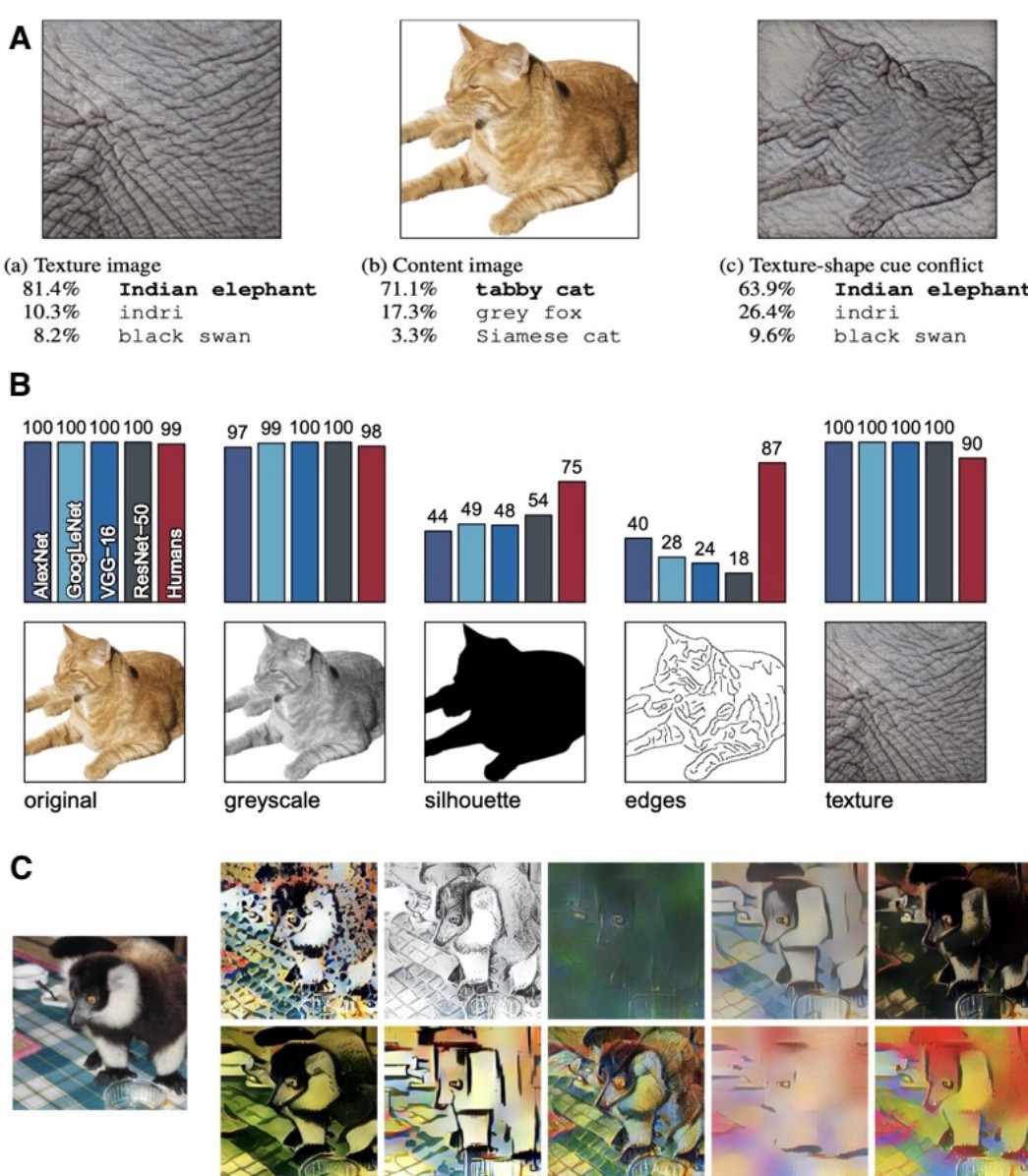

Figure 13: A) Classification of a standard ResNet-50 of (a) a texture image (elephant skin: only texture cues); (b) a normal image of a cat (with both shape and texture cues), and (c) an image with a texture-shape cue conflict, generated by style transfer between the first two images, B) Accuracy and example stimuli for five different experiments without cue conflict, and C) Sample images from the Stylized-ImageNet (SIN) dataset created by applying AdaIN style transfer to an ImageNet image (left). Figure compiled from Geirhos et al. (2018).

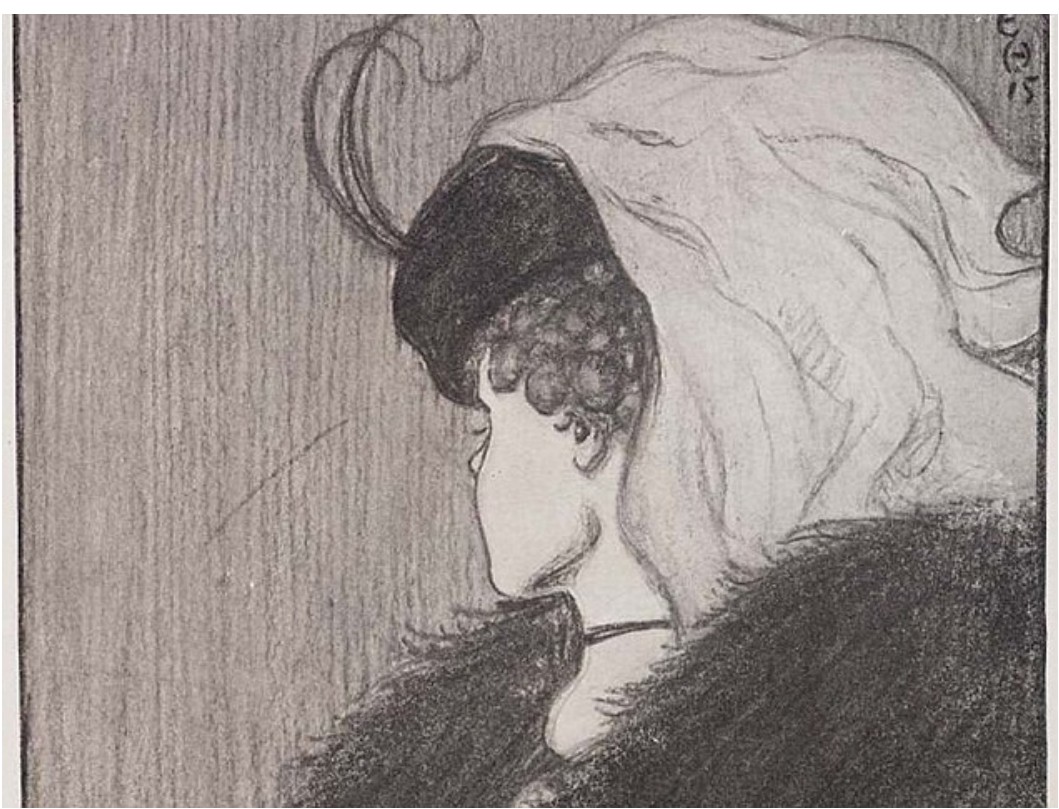

Figure 14: An example visual illusion simultaneously depicting a portrait of a young lady or an old lady. While fooling humans takes a lot of effort and special skills are needed, deep models are much easier to be fooled. In this example, the artist has carefully added features to make the portrait look like an old lady while the new additions will not negatively impact the look of the young lady too much. For example, the right eyebrow of the old lady (marked in red below) does not distort the ear of the young lady too much. See `https://medium.com/@jonathan_hui/adversarial-attacks-b58318bb497b` for more details.

| Display Image | Shape-Object | Texture-Object | 1st Choice | 2nd Choice | 3rd Choice | 4th Choice | 5th Choice |
|---|---|---|---|---|---|---|---|
| | Otter (0%) | Odometer (0.03%) | Can Opener (12.12%) | Electric Guitar (7.66%) | Hook (3.64%) | Remote Control (3.54%) | Corkscrew (3.53%) |
| | Ram (31.8%) | Bison (0.06%) | Ibex (46.51%) | Ram (31.8%) | Bighorn (18.1%) | Chesapeake Bay Retriever (1.09%) | Hyena (0.48%) |
| | Sturgeon (0.02%) | Honeycomb (0.48%) | Starfish (15.36%) | Banded Gecko (5.6%) | Electric Ray (5.57%) | Snail (5.56%) | Kite (4.52%) |
| | Bee (0.04%) | Velvet (6.97%) | Stole (29.94%) | Wool (10.24%) | Velvet (6.97%) | Bonnet (4.53%) | Poncho (4.2%) |
| | Bison (0.01%) | Stone Wall (8.72%) | Stone Wall (8.72%) | Parachute (7.9%) | Tile Roof (5.19%) | Stole (3.6%) | Kite (3.26%) |
| | Dugong (0.03%) | Gorilla (0%) | Bluetick (8.53%) | German Short-Haired Pointer (7.73%) | Egyptian Cat (6.19%) | Tabby (5.79%) | Kerry Blue Terrier (5.6%) |
| | Flamingo (0.83%) | Python (0.05%) | Limpkin (13.48%) | Hook (12.46%) | Corkscrew (6.33%) | Bustard (3.8%) | Ostrich (3.63%) |
| | Gorilla (0%) | Green Mamba (1.48%) | Strawberry (19.03%) | Custard Apple (16.42%) | Granny Smith (6.81%) | Bell Pepper (5.47%) | Knot (4.5%) |
| | Hyena (0.15%) | Cask (0.19%) | Triceratops (19.8%) | Mortarboard (3.2%) | Pedestal (2.4%) | Pencil Sharpener (2.02%) | Kite (1.78%) |
| | Lobster (0.02%) | Broom (0.83%) | Wool (18.69%) | Rocking Chair (4.8%) | Knot (4.67%) | Hamper (4.41%) | Stole (3.36%) |

Figure 15: Classification results based on shape vs. texture. The left-most column shows the image presented to a model. The second column in each row names the object from which the shape was sampled. The third column names the object from which the textured silhouette was obtained. Probabilities assigned to the object name in columns 2 and 3 are shown as percents below the object label. The remaining five columns show the probabilities (as percents) produced by the network for its top five classifications, ordered left to right in terms of probability. Correct shape classifications in the top five are shaded in blue and correct texture classifications are shaded in orange. Figure from Baker et al. (2018).

## B  ADDITIONAL RESULTS FOR THE EDGE-AUGMENTED DEFENSE

Results of the shape defense (Algorithm 1 in the main text) over eight datasets. Tables with (*) in their caption have contributed to Fig 3 in the main text. In some tables, results are computed when the edge map is computed from all image channels, i.e., a gray-level image is first computed by averaging the 4 image channels (Img + Edge map) and then a new edge map is derived.

Table 4: Results over the Fashion MNIST dataset (*)

| | Orig. model | | | | Rob. model (8) | | Rob. model (32) | | Rob. model (64) | | Average |
|---|---|---|---|---|---|---|---|---|---|---|---|
| $\epsilon$ | 0/clean | 8 | 32 | 64 | 0/clean | 8 | 0/clean | 32 | 0/clean | 64 | Rob. models |
| **FGSM** | | | | | | | | | | | |
| Edge | 0.775 | 0.714 | 0.497 | 0.089 | 0.776 | 0.740 | 0.766 | **0.664** | 0.748 | **0.750** | **0.741** |
| Img2Edge | ,, | **0.755** | **0.679** | **0.452** | ,, | **0.762** | ,, | **0.664** | ,, | 0.420 | 0.690 |
| Img | 0.798 | 0.670 | 0.288 | 0.027 | **0.798** | 0.722 | 0.764 | 0.584 | **0.768** | 0.505 | 0.690 |
| Img+Edge | **0.809** | 0.662 | 0.229 | 0.010 | 0.794 | 0.732 | 0.769 | 0.623 | 0.750 | 0.537 | 0.701 |
| Redetect | ,, | 0.691 | 0.326 | 0.053 | ,, | 0.739 (0.761) | ,, | 0.616 (0.660) | ,, | 0.491 (0.496) | 0.693 |
| **Img + Redetected Edge** | | | | | 0.789 | 0.719 | **0.775** | 0.539 | 0.762 | 0.045 | 0.605 |
| Redetect | | | | | ,, | 0.739 (0.753) | ,, | **0.664** (0.678) | ,, | 0.611 (0.532) | 0.721 |
| **PGD-40** | | | | | | | | | | | |
| Edge | 0.775 | 0.711 | 0.370 | 0.002 | 0.783 | 0.744 | 0.769 | 0.661 | 0.743 | 0.574 | 0.712 |
| Img2Edge | ,, | **0.757** | **0.683** | **0.380** | ,, | **0.762** | ,, | 0.658 | ,, | 0.374 | 0.681 |
| Img | 0.798 | 0.659 | 0.133 | 0.000 | 0.792 | 0.713 | 0.760 | 0.515 | 0.734 | 0.324 | 0.640 |
| Img+Edge | **0.809** | 0.647 | 0.100 | 0.000 | 0.794 | 0.726 | 0.765 | 0.608 | 0.744 | 0.568 | 0.701 |
| Redetect | ,, | 0.682 | 0.235 | 0.014 | ,, | 0.734 (0.760) | ,, | 0.629 (0.666) | - | 0.607 (0.426) | 0.712 |
| **Img + Redetected Edge** | | | | | **0.800** | 0.717 | **0.779** | 0.393 | **0.771** | 0.002 | 0.577 |
| Redetect | | | | | ,, | 0.743 (0.766) | ,, | **0.694** (0.681) | ,, | **0.690** (0.504) | **0.746** |

Table 5: Results over the TinyImageNet dataset (*)

| | Orig. model | | | Rob. model (8) | | Rob. model (32) | | Average |
|---|---|---|---|---|---|---|---|---|
| $\epsilon$ | 0/clean | 8 | 32 | 0/clean | 8 | 0/clean | 32 | Rob. models |
| **FGSM** | | | | | | | | |
| Edge | 0.136 | 0.010 | 0.001 | 0.150 | 0.078 | 0.098 | 0.021 | 0.087 |
| Img2Edge | ,, | 0.097 | **0.096** | ,, | 0.094 | ,, | 0.077 | 0.105 |
| Img | **0.531** | 0.166 | 0.074 | **0.512** | 0.297 | **0.488** | 0.168 | **0.366** |
| Img + Edge | 0.522 | 0.152 | 0.050 | 0.508 | 0.273 | 0.471 | 0.148 | 0.350 |
| Redetect | ,, | **0.171** | 0.081 | ,, | 0.287 (0.356) | ,, | 0.162 (0.266) | 0.357 |
| **Img + Redetected Edge** | | | | 0.505 | 0.264 | 0.482 | 0.111 | 0.340 |
| Redetect | | | | ,, | **0.305** (0.371) | ,, | **0.171** (0.296) | **0.366** |
| **PGD-40** | | | | | | | | |
| Edge | 0.136 | 0.007 | 0.000 | 0.148 | 0.077 | 0.039 | 0.014 | 0.069 |
| Img2Edge | ,, | **0.094** | **0.092** | ,, | 0.095 | ,, | 0.033 | 0.079 |
| Img | **0.531** | 0.019 | 0.000 | 0.392 | 0.150 | 0.191 | 0.019 | 0.188 |
| Img + Edge | 0.522 | 0.008 | 0.000 | 0.402 | 0.131 | 0.157 | 0.003 | 0.173 |
| Redetect | ,, | 0.074 | 0.009 | ,, | 0.198 (0.353) | ,, | 0.019 (0.103) | 0.194 |
| **Img + Redetected Edge** | | | | **0.425** | 0.072 | **0.328** | 0.005 | 0.208 |
| Redetect | | | | ,, | **0.206** (0.380) | ,, | **0.073** (0.279) | **0.258** |

Table 6: Results on CIFAR-10 dataset [edge map computed from 4 channels]

| | Orig. model | | | Rob. model (8) | | Rob. model (32) | | Average |
|---|---|---|---|---|---|---|---|---|
| $\epsilon$ | 0/clean | 8 | 32 | 0/clean | 8 | 0/clean | 32 | **Rob. models** |
| **FGSM** | | | | | | | | |
| **Img+Edge** | 0.860 | 0.366 | 0.169 | 0.846 | 0.611 | 0.815 | 0.442 | 0.679 |
| Redetect | " | 0.415 | 0.280 | " | 0.574 | ,, | 0.416 | 0.663 |
| **Img + Redetected Edge** | | | | 0.848 | 0.547 | 0.835 | 0.351 | 0.645 |
| Redetect | | | | " | 0.696 | " | 0.553 | 0.733 |
| **PGD-40** | | | | | | | | |
| **Img+Edge** | 0.860 | | 0.000 | 0.789 | 0.431 | 0.179 | 0.135 | 0.384 |
| Redetect | " | | 0.087 | ,, | 0.501 | ,, | 0.152 | 0.405 |
| **Img + Redetected Edge** | | | | 0.837 | 0.164 | 0.767 | 0.010 | 0.444 |
| Redetect | | | | " | 0.648 | ,, | 0.352 | 0.651 |

Table 7: Results on CIFAR dataset using Sobel edge detection [edge map computed from 4 channels]

| | Orig. model | | | Rob. model (8) | | Rob. model (32) | | Average |
|---|---|---|---|---|---|---|---|---|
| $\epsilon$ | 0/clean | 8 | 32 | 0/clean | 8 | 0/clean | 32 | **Rob. models** |
| **FGSM** | | | | | | | | |
| **Img+Edge** | 0.876 | 0.331 | 0.207 | 0.856 | 0.613 | 0.829 | 0.469 | 0.692 |
| Redetect | ,, | 0.424 | 0.285 | ,, | 0.645 | ,, | 0.490 | 0.705 |
| **Img + Redetected Edge** | | | | 0.858 | 0.580 | 0.842 | 0.411 | 0.673 |
| Redetect | | | | " | 0.685 | ,, | 0.558 | 0.736 |

Table 8: Results on DogVsCat dataset [edge map computed from 4 channels] (*)

| | Orig. model | | | Rob. model (8) | | Rob. model (32) | | Average |
|---|---|---|---|---|---|---|---|---|
| $\epsilon$ | 0/clean | 8 | 32 | 0/clean | 8 | 0/clean | 32 | **Rob. models** |
| **FGSM** | | | | | | | | |
| **Edge** | 0.814 | 0.633 | 0.119 | 0.812 | 0.757 | 0.806 | **0.999** | 0.843 |
| **Img2Edge** | ,, | **0.755** | **0.584** | ,, | **0.767** | ,, | 0.576 | 0.740 |
| **Img** | **0.863** | 0.007 | 0.051 | 0.777 | 0.430 | **0.819** | 0.985 | 0.753 |
| **Img+Edge** | 0.823 | 0.007 | 0.000 | 0.782 | 0.641 | 0.808 | 0.992 | 0.806 |
| Redetect | " | 0.043 | 0.002 | " | 0.666 | " | 0.986 | 0.810 |
| **Img + Redetected Edge** | | | | **0.829** | 0.615 | 0.812 | 0.853 | 0.778 |
| Redetect | | | | " | 0.763 | " | 0.998 | **0.850** |
| **PGD-40** | | | | | | | | |
| **Edge** | 0.814 | 0.624 | 0.018 | **0.820** | 0.770 | 0.763 | 0.681 | 0.758 |
| **Img2Edge** | ,, | 0.760 | 0.568 | ,, | **0.778** | ,, | 0.656 | 0.754 |
| **Img** | **0.863** | 0.000 | 0.000 | 0.769 | 0.384 | 0.500 | 0.500 | 0.538 |
| **Img+Edge** | 0.823 | 0.000 | 0.000 | 0.785 | 0.689 | 0.816 | 0.496 | 0.696 |
| Redetect | " | 0.006 | 0.000 | ,, | 0.744 | ,, | 0.500 | 0.711 |
| **Img + Redetected Edge** | | | | 0.819 | 0.600 | **0.817** | 0.009 | 0.561 |
| Redetect | | | | " | 0.760 | ,, | **0.972** | **0.842** |

Table 9: Results on DogBreeds dataset using Sobel edge detection [edge map computed from 4 channels] (*)

| | Orig. model | | | Rob. model (8) | | Rob. model (32) | | Average Rob. models |
|---|---|---|---|---|---|---|---|---|
| $\epsilon$ | 0/clean | 8 | 32 | 0/clean | 8 | 0/clean | 32 | |
| | | | | **FGSM** | | | | |
| Edge | 0.750 | 0.006 | 0.031 | 0.506 | 0.101 | 0.413 | 0.073 | 0.273 |
| Img2Edge | ,, | 0.236 | **0.194** | ,, | 0.362 | ,, | 0.241 | 0.380 |
| Img | **0.899** | 0.256 | 0.140 | 0.823 | 0.595 | 0.829 | **0.449** | 0.674 |
| Img + Edge | 0.896 | 0.225 | 0.098 | **0.862** | 0.534 | 0.820 | 0.385 | 0.650 |
| Redetect | ,, | **0.244** | 0.171 | ,, | 0.455 | ,, | 0.292 | 0.607 |
| Img + Redetected Edge | | | | 0.843 | 0.506 | **0.874** | 0.298 | 0.630 |
| Redetect | | | | " | **0.618** | ,, | 0.419 | **0.689** |
| | | | | **PGD-40** | | | | |
| Edge | 0.750 | 0.000 | 0.000 | 0.514 | 0.065 | 0.036 | 0.000 | 0.154 |
| Img2Edge | ,, | **0.250** | **0.207** | ,, | 0.301 | ,, | 0.037 | 0.222 |
| Img | **0.899** | 0.000 | 0.000 | 0.795 | 0.286 | 0.596 | 0.025 | 0.425 |
| Img + Edge | 0.896 | 0.000 | 0.000 | 0.789 | 0.225 | 0.567 | 0.042 | 0.406 |
| Redetect | ,, | 0.008 | 0.000 | " | 0.396 | ,, | 0.065 | 0.454 |
| Img + Redetected Edge | | | | 0.772 | 0.028 | **0.677** | 0.000 | 0.369 |
| Redetect | | | | " | **0.393** | " | **0.149** | **0.498** |

Table 10: Results on DogBreeds dataset using Sobel edge detection [edge map computed from 3 channels]

| | Orig. model | | | Rob. model (8) | | Rob. model (32) | | Average Rob. models |
|---|---|---|---|---|---|---|---|---|
| $\epsilon$ | 0/clean | 8 | 32 | 0/clean | 8 | 0/clean | 32 | |
| | | | | **FGSM** | | | | |
| Img + Edge | 0.888 | 0.177 | 0.073 | 0.882 | 0.455 | 0.812 | 0.261 | 0.602 |
| Redetect | ,, | 0.258 | 0.110 | ,, | 0.502 | ,, | 0.275 | 0.618 |
| Img + Redetected Edge | | | | 0.893 | 0.480 | 0.848 | 0.216 | 0.609 |
| Redetect | | | | ,, | 0.626 | ,, | 0.388 | 0.689 |

Table 11: Results on GTSRB dataset [edge map computed from 4 channels] (*)

| | Orig. model | | | Rob. model (8) | | Rob. model (32) | | Average Rob. models |
|---|---|---|---|---|---|---|---|---|
| $\epsilon$ | 0/clean | 8 | 32 | 0/clean | 8 | 0/clean | 32 | |
| | | | | **FGSM** | | | | |
| Edge | 0.938 | **0.683** | 0.315 | **0.947** | **0.863** | **0.946** | 0.701 | 0.864 |
| Img2Edge | ,, | 0.501 | 0.451 | ,, | 0.516 | ,, | 0.469 | 0.719 |
| Img | **0.955** | 0.464 | 0.322 | 0.902 | 0.607 | 0.896 | 0.562 | 0.742 |
| Img + Edge | 0.951 | 0.624 | 0.382 | 0.940 | 0.842 | 0.943 | 0.686 | 0.853 |
| Redetect | " | 0.592 | **0.471** | " | 0.743 | " | 0.626 | 0.813 |
| Img + Redetected Edge | | | | 0.925 | 0.801 | 0.939 | 0.616 | 0.820 |
| Redetect | | | | " | 0.844 | " | **0.766** | **0.869** |
| | | | | **PGD-40** | | | | |
| Edge | 0.938 | **0.618** | 0.054 | **0.950** | **0.861** | **0.937** | **0.598** | **0.836** |
| Img2Edge | ,, | 0.501 | **0.459** | ,, | 0.506 | ,, | 0.462 | 0.714 |
| Img | **0.955** | 0.189 | 0.033 | 0.855 | 0.495 | 0.736 | 0.246 | 0.583 |
| Img + Edge | 0.951 | 0.271 | 0.021 | 0.943 | 0.750 | 0.839 | 0.342 | 0.718 |
| Redetect | ,, | 0.526 | 0.251 | ,, | 0.774 | ,, | 0.514 | 0.767 |
| Img + Redetected Edge | | | | 0.929 | 0.505 | 0.893 | 0.134 | 0.615 |
| Redetect | | | | " | 0.818 | ,, | 0.557 | 0.799 |

Table 12: Results on GTSRB dataset [edge map computed from 3 channels]

| | Orig. model | | | Rob. model (8) | | Rob. model (32) | | Average Rob. models |
|---|---|---|---|---|---|---|---|---|
| $\epsilon$ | 0/clean | 8 | 32 | 0/clean | 8 | 0/clean | 32 | |
| | | | | **FGSM** | | | | |
| Img + Edge | 0.951 | 0.624 | 0.382 | 0.940 | 0.842 | 0.943 | 0.686 | 0.853 |
| Redetect | ,, | 0.500 | 0.395 | ,, | 0.558 | ,, | 0.492 | 0.733 |
| Img + Redetected Edge | | | | 0.889 | 0.699 | 0.891 | 0.549 | 0.757 |
| Redetect | | | | ,, | 0.610 | ,, | 0.577 | 0.742 |

Table 13: Results on Icons-50 dataset [edge map computed from 4 channels] (*)

| | Orig. model | | | Rob. model (8) | | Rob. model (32) | | Average |
|---|---|---|---|---|---|---|---|---|
| $\epsilon$ | 0/clean | 8 | 32 | 0/clean | 8 | 0/clean | 32 | Rob. models |
| **FGSM** | | | | | | | | |
| **Edge** | 0.883 | 0.545 | 0.210 | **0.904** | 0.771 | **0.889** | 0.594 | 0.789 |
| **Img2Edge** | ,, | **0.713** | **0.690** | ,, | 0.746 | ,, | 0.730 | **0.817** |
| **Img** | **0.930** | 0.495 | 0.433 | 0.772 | 0.789 | 0.836 | 0.720 | 0.779 |
| **Img + Edge** | 0.929 | 0.569 | 0.433 | 0.829 | 0.818 | 0.844 | **0.745** | 0.809 |
| Redetect | ,, | 0.470 | 0.414 | ,, | 0.730 | ,, | 0.732 | 0.784 |
| **Img + Redetected Edge** | | | | 0.841 | **0.837** | 0.849 | 0.688 | 0.804 |
| Redetect | | | | ,, | 0.817 | ,, | 0.710 | 0.804 |
| **PGD-40** | | | | | | | | |
| **Edge** | 0.883 | 0.423 | 0.000 | **0.902** | 0.769 | **0.846** | 0.404 | 0.730 |
| **Img2Edge** | ,, | **0.706** | **0.683** | ,, | 0.753 | ,, | **0.695** | **0.799** |
| **Img** | **0.930** | 0.341 | 0.113 | 0.765 | 0.663 | 0.736 | 0.453 | 0.654 |
| **Img + Edge** | 0.929 | 0.320 | 0.011 | 0.800 | 0.678 | 0.785 | 0.366 | 0.657 |
| Redetect | ,, | 0.416 | 0.248 | ,, | 0.738 | ,, | 0.660 | 0.746 |
| **Img + Redetected Edge** | | | | 0.838 | 0.644 | 0.824 | 0.097 | 0.601 |
| Redetect | | | | " | **0.792** | ,, | 0.539 | 0.748 |

Table 14: Results on Icons-50 dataset [edge map computed from 3 channels]

| | Orig. model | | | Rob. model (8) | | Rob. model (32) | | Average |
|---|---|---|---|---|---|---|---|---|
| $\epsilon$ | 0/clean | 8 | 32 | 0/clean | 8 | 0/clean | 32 | Rob. models |
| **FGSM** | | | | | | | | |
| **Img+Edge** | 0.929 | 0.569 | 0.433 | 0.829 | 0.818 | 0.844 | 0.745 | 0.809 |
| Redetect | ,, | 0.520 | 0.460 | ,, | 0.737 | ,, | 0.731 | 0.785 |
| **Img + Redetected Edge** | | | | 0.831 | 0.788 | 0.870 | 0.725 | 0.804 |
| Redetect | | | | " | 0.783 | ,, | 0.765 | 0.812 |

Table 15: Results on Sketch dataset [edge map computed from 2 channels] (*)

| | Orig. model | | | Rob. model (8) | | Rob. model (32) | | Average |
|---|---|---|---|---|---|---|---|---|
| $\epsilon$ | 0/clean | 8 | 32 | 0/clean | 8 | 0/clean | 32 | Rob. models |
| **FGSM** | | | | | | | | |
| **Edge** | 0.479 | 0.167 | **0.041** | 0.502 | 0.343 | **0.483** | **0.216** | **0.386** |
| **Img2Edge** | ,, | **0.464** | 0.014 | ,, | **0.494** | ,, | 0.022 | 0.375 |
| **Img** | **0.532** | 0.109 | 0.278 | **0.530** | 0.278 | 0.474 | 0.144 | 0.356 |
| **Gray + Edge** | 0.486 | 0.097 | 0.019 | 0.513 | 0.286 | 0.440 | 0.167 | 0.352 |
| Redetect | ,, | 0.263 | 0.004 | ,, | 0.355 | ,, | 0.013 | 0.330 |
| **Img + Redetected Edge** | | | | 0.497 | 0.180 | 0.420 | 0.071 | 0.292 |
| Redetect | | | | " | 0.416 | ,, | 0.162 | 0.374 |
| **PGD-40** | | | | | | | | |
| **Edge** | 0.480 | 0.106 | 0.000 | 0.508 | 0.341 | 0.401 | 0.068 | 0.330 |
| **Img2Edge** | ,, | **0.471** | **0.127** | ,, | **0.499** | ,, | **0.214** | **0.405** |
| **Img** | **0.532** | 0.028 | 0.000 | **0.538** | 0.260 | 0.018 | 0.000 | 0.204 |
| **Gray + Edge** | 0.486 | 0.034 | 0.000 | 0.500 | 0.279 | 0.026 | 0.000 | 0.201 |
| Redetect | ,, | 0.277 | 0.024 | ,, | 0.360 | ,, | 0.004 | 0.223 |
| **Img + Redetected Edge** | | | | 0.502 | 0.121 | **0.448** | 0.000 | 0.268 |
| Redetect | | | | " | 0.423 | ,, | 0.212 | 0.396 |

Table 16: Results on Sketch dataset [edge map computed from 1 channel]

| | Orig. model | | | Rob. model (8) | | Rob. model (32) | | Average |
|---|---|---|---|---|---|---|---|---|
| $\epsilon$ | 0/clean | 8 | 32 | 0/clean | 8 | 0/clean | 32 | Rob. models |
| **FGSM** | | | | | | | | |
| **Gray + Edge** | 0.486 | 0.097 | 0.019 | 0.513 | 0.286 | 0.440 | 0.167 | 0.352 |
| Redetect | ,, | 0.213 | 0.005 | ,, | 0.388 | ,, | 0.022 | 0.341 |
| **Img + Redetected Edge** | | | | 0.519 | 0.296 | 0.445 | 0.191 | 0.363 |
| Redetect | | | | ,, | 0.397 | ,, | 0.020 | 0.345 |

Table 17: Results on Imagenette2-160 dataset [edge map computed from 4 channels] (*)

| | Orig. model | | | Rob. model (8) | | Rob. model (32) | | Average Rob. models |
|---|---|---|---|---|---|---|---|---|
| $\epsilon$ | 0/clean | 8 | 32 | 0/clean | 8 | 0/clean | 32 | |
| **FGSM** | | | | | | | | |
| **Edge** | 0.780 | 0.101 | 0.436 | 0.781 | 0.520 | 0.664 | 0.245 | 0.553 |
| **Img2Edge** | ,, | 0.599 | **0.598** | ,, | 0.603 | ,, | 0.578 | 0.656 |
| **Img** | **0.969** | 0.617 | 0.409 | **0.959** | 0.827 | 0.946 | 0.710 | 0.860 |
| **Img + Edge** | 0.959 | 0.613 | 0.373 | 0.951 | 0.801 | 0.935 | 0.643 | 0.832 |
| Redetect | ,, | **0.652** | 0.471 | ,, | 0.812 | ,, | 0.687 | 0.846 |
| **Img + Redetected Edge** | | | | 0.950 | 0.747 | **0.949** | 0.592 | 0.810 |
| Redetect | | | | ,, | **0.834** | ,, | **0.732** | **0.866** |
| **PGD-40** | | | | | | | | |
| **Edge** | 0.780 | 0.064 | 0.000 | 0.794 | 0.526 | 0.577 | 0.071 | 0.492 |
| **Img2Edge** | ,, | **0.601** | **0.577** | ,, | 0.610 | ,, | 0.381 | 0.591 |
| **Img** | **0.969** | 0.052 | 0.005 | 0.918 | 0.599 | 0.808 | 0.221 | 0.636 |
| **Img + Edge** | 0.959 | 0.045 | 0.000 | 0.909 | 0.558 | 0.762 | 0.151 | 0.595 |
| Redetect | " | 0.445 | 0.069 | " | 0.743 | " | 0.305 | 0.680 |
| **Img + Redetected Edge** | | | | **0.944** | 0.246 | **0.883** | 0.046 | 0.530 |
| Redetect | | | | " | **0.757** | " | **0.432** | **0.754** |

Table 18: Results on Imagenette2-160 dataset [edge map computed from 3 channels]

| | Orig. model | | | Rob. model (8) | | Rob. model (32) | | Average Rob. models |
|---|---|---|---|---|---|---|---|---|
| $\epsilon$ | 0/clean | 8 | 32 | 0/clean | 8 | 0/clean | 32 | |
| **FGSM** | | | | | | | | |
| **Img + Edge** | 0.959 | 0.613 | 0.373 | 0.951 | 0.801 | 0.935 | 0.643 | 0.833 |
| Redetect | ,, | 0.611 | 0.447 | ,, | 0.802 | ,, | 0.673 | 0.840 |
| **Img + Redetected Edge** | | | | 0.952 | 0.767 | 0.949 | 0.596 | 0.816 |
| Redetect | | | | ,, | 0.832 | ,, | 0.729 | 0.865 |

## C  SUMMARY RESULTS OF USING EDGES FOR ADVERSARIAL DEFENSE

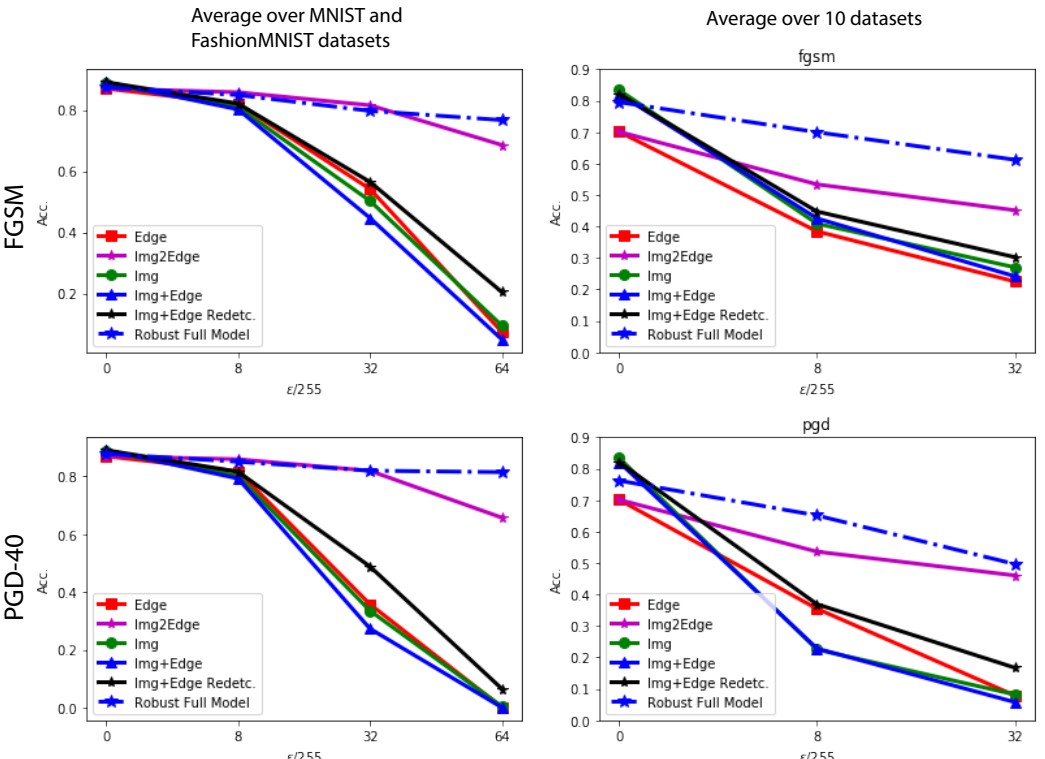

Figure 16: Comparison of natural (the Orig. column in the tables; solid curves) *vs.* adversarial training (blue dashed-dot curves). The accuracy at $\epsilon = 0$ (for adversarial training) is averaged over different robust models (three over MNIST and two over others; corresponding to clean columns in tables). Left column) Average over MNIST and Fashion MNIST datasets, Right column) Average over all datasets. Results show a clear advantage of using edges. Over MNIST and FashionMNIST, the model trained on edges alone leads to a trade-off between accuracy and robustness. Img+edge model does worse than the Image model but its performance is recovered after adversarial training. Img2Edge model wins over models using natural training. Please see also tables in the main text and Appx. B and the explanation in the main text. Overall, incorporating edge and image together and redetection at inference times leads to higher accuracy and robustness.

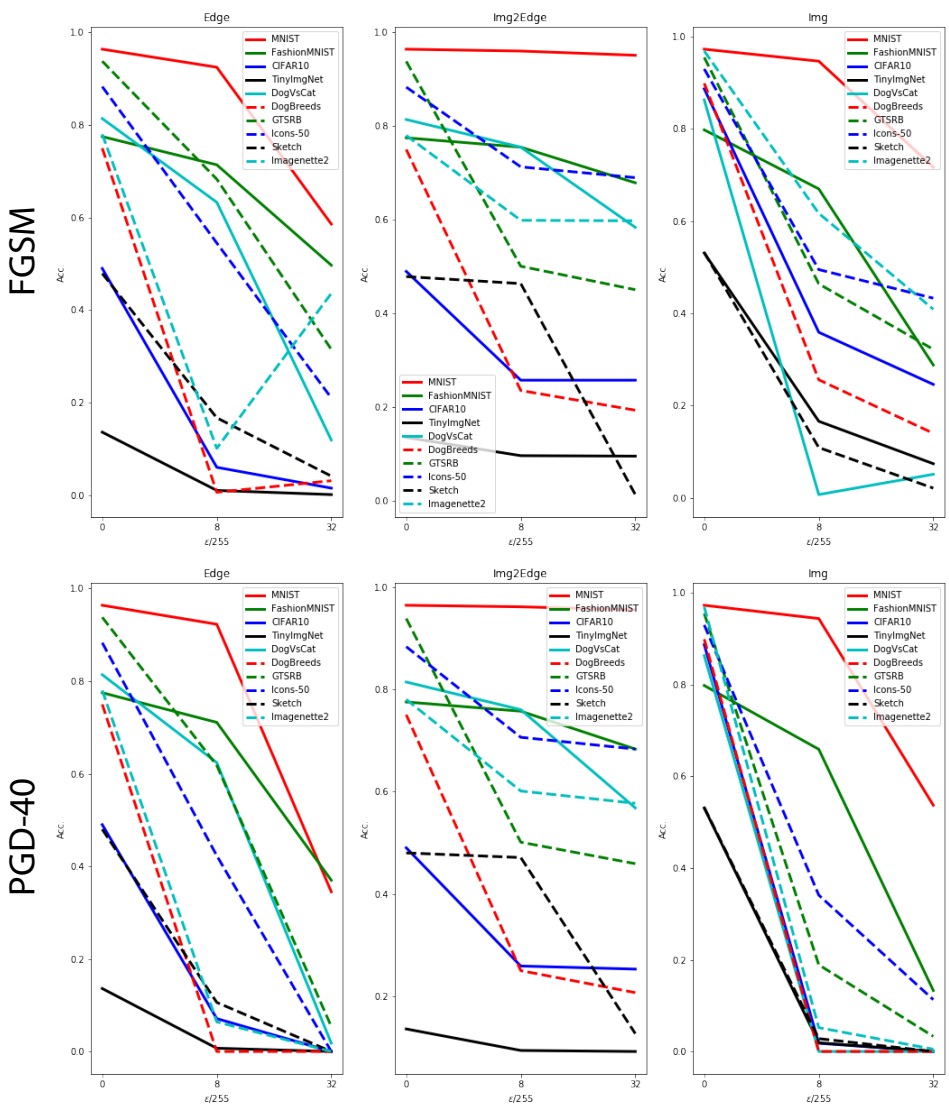

Figure 17: Breakdown of natural training (the Orig. row in Tables) over datasets.

# D ANALYSIS OF PARAMETER $\alpha$ IN ALG. 1 (EAT DEFENSE)

Table 19: Results (Top-1 acc.) over MNIST corresponding to $\alpha = 0$ (i.e., adversarial training only on adversarial examples taking part in the loss function). See also Table 1 in the main text.

| | Rob. model (8) | | Rob. model (32) | | Rob. model (64) | | Average |
|---|---|---|---|---|---|---|---|
| $\epsilon$ | 0/clean | 8 | 0/clean | 32 | 0/clean | 64 | Rob. models |
| **FGSM** | | | | | | | |
| **Img+Edge** | **0.963** | 0.938 | **0.959** | 0.869 | 0.931 | 0.684 | 0.891 |
| Redetect | ,, | 0.943 | ,, | 0.887 | ,, | 0.727 | 0.902 |
| **Img + Redetected Edge** | **0.963** | 0.936 | 0.944 | 0.588 | **0.937** | 0.030 | 0.733 |
| Redetect | ,, | **0.948** | ,, | **0.911** | ,, | **0.916** | **0.937** |
| **PGD-40** | | | | | | | |
| **Img+Edge** | **0.966** | 0.940 | **0.960** | 0.859 | 0.928 | 0.607 | 0.877 |
| Redetect | ,, | **0.946** | ,, | 0.883 | ,, | 0.657 | 0.890 |
| **Img + Redetected Edge** | 0.963 | 0.933 | 0.947 | 0.469 | **0.936** | 0.000 | 0.708 |
| Redetect | ,, | **0.946** | ,, | **0.913** | ,, | **0.915** | **0.937** |

Table 20: Results (Top-1 acc.) over Fashion MNIST corresponding to $\alpha = 0$ (i.e., adversarial training only on adversarial examples taking part in the loss function). See also Table 4 in the main text.

| | Rob. model (8) | | Rob. model (32) | | Rob. model (64) | | Average |
|---|---|---|---|---|---|---|---|
| $\epsilon$ | 0/clean | 8 | 0/clean | 32 | 0/clean | 64 | Rob. models |
| **FGSM** | | | | | | | |
| **Img+Edge** | 0.756 | 0.701 | 0.732 | 0.619 | 0.683 | 0.487 | 0.663 |
| Redetect | ,, | 0.707 | ,, | 0.635 | ,, | 0.481 | 0.666 |
| **Img + Redetected Edge** | **0.768** | 0.705 | **0.739** | 0.481 | **0.693** | 0.040 | 0.571 |
| Redetect | ,, | **0.727** | ,, | **0.660** | ,, | **0.635** | **0.704** |
| **PGD-40** | | | | | | | |
| **Img+Edge** | 0.768 | 0.702 | 0.749 | 0.573 | 0.718 | 0.432 | 0.657 |
| Redetect | ,, | 0.714 | ,, | 0.593 | ,, | 0.510 | 0.675 |
| **Img + Redetected Edge** | **0.778** | 0.702 | **0.762** | 0.414 | **0.750** | 0.001 | 0.568 |
| Redetect | ,, | **0.725** | ,, | **0.632** | ,, | **0.615** | **0.710** |

# E   RESULTS OF THE SUBSTITUTE MODEL ATTACK

Table 21: Results of the substitute attack against the robust Img + Edge models (redetect and full model).

| | MNIST | | | Fashion MNIST | | | CIFAR | | TinyImgNet | |
|---|---|---|---|---|---|---|---|---|---|---|
| $\epsilon$ | 8 | 32 | 64 | 8 | 32 | 64 | 8 | 32 | 8 | 32 |

**FGSM**

**Img + edge model (redetect inference)**

| | | | | | | | | | | |
|---|---|---|---|---|---|---|---|---|---|---|
| Substitute model on clean images | 0.94 | 0.9365 | 0.9314 | 0.7515 | 0.7393 | 0.7311 | 0.8079 | 0.7766 | 0.008 | 0.008 |
| Substitute model on adversarial images | 0.8941 | 0.5858 | 0.0992 | 0.6484 | 0.3701 | 0.0967 | 0.2716 | 0.2049 | 0.004 | 0.003 |
| Robust model on clean images | 0.9761 | 0.9766 | 0.9722 | 0.7939 | 0.7692 | 0.75 | 0.8463 | 0.8463 | 0.508 | 0.471 |
| Robust model on adversarial images | 0.9623 | 0.9189 | 0.842 | 0.7391 | 0.6156 | 0.4908 | 0.5695 | 0.4186 | 0.287 | 0.161 |
| Robust model on substitute adv. images | 0.9678 | 0.9472 | 0.8813 | 0.7609 | 0.6604 | 0.4955 | 0.6307 | 0.5463 | 0.356 | 0.266 |

**Img + redetected edge model (redetect inference)**

| | | | | | | | | | | |
|---|---|---|---|---|---|---|---|---|---|---|
| Substitute model on clean images | 0.9381 | 0.9335 | 0.9326 | 0.7513 | 0.7431 | 0.7388 | 0.8104 | 0.7966 | 0.008 | 0.008 |
| Substitute model on adversarial images | 0.89 | 0.5696 | 0.0989 | 0.6538 | 0.3663 | 0.08 | 0.2879 | 0.1988 | 0.004 | 0.002 |
| Robust model on clean images | 0.9742 | 0.9699 | 0.9681 | 0.7891 | 0.7746 | 0.7617 | 0.8456 | 0.8328 | 0.495 | 0.482 |
| Robust model on adversarial images | 0.9583 | 0.9283 | 0.9216 | 0.7392 | 0.664 | 0.6115 | 0.7032 | 0.5684 | 0.380 | 0.170 |
| Robust model on substitute adv. images | 0.9657 | 0.9469 | 0.9249 | 0.7529 | 0.6776 | 0.5318 | 0.7528 | 0.7528 | 0.371 | 0.296 |

**PGD-40**

**Img + edge model (redetect inference)**

| | | | | | | | | | | |
|---|---|---|---|---|---|---|---|---|---|---|
| Substitute model on clean images | 0.9391 | 0.9344 | 0.9257 | 0.7531 | 0.7408 | 0.7303 | 0.756 | 0.194 | 0.008 | 0.006 |
| Substitute model on adv. images | 0.8906 | 0.4455 | 0.0196 | 0.6473 | 0.2745 | 0.0096 | 0.020 | 0.003 | 0.000 | 0.000 |
| Robust model on clean images | 0.9782 | 0.9751 | 0.9654 | 0.7938 | 0.7652 | 0.7442 | 0.788 | 0.179 | 0.395 | 0.157 |
| Robust model on adv. images | 0.9599 | 0.9132 | 0.8039 | 0.7336 | 0.6289 | 0.6068 | 0.504 | 0.152 | 0.242 | 0.018 |
| Robust model on substitute adv. images | 0.9667 | 0.9477 | 0.9079 | 0.7603 | 0.6656 | 0.4263 | 0.646 | 0.170 | 0.352 | 0.103 |

**Img + redetected edge model (redetect inference)**

| | | | | | | | | | | |
|---|---|---|---|---|---|---|---|---|---|---|
| Substitute model on clean images | 0.9385 | 0.9363 | 0.9329 | 0.7503 | 0.7471 | 0.7415 | 0.804 | 0.730 | 0.008 | 0.008 |
| Substitute model on adv. images | 0.8888 | 0.4617 | 0.0211 | 0.6458 | 0.2687 | 0.01 | 0.016 | 0.000 | 0.000 | 0.000 |
| Robust model on clean images | 0.975 | 0.9732 | 0.9682 | 0.7998 | 0.7793 | 0.7715 | 0.834 | 0.766 | 0.425 | 0.328 |
| Robust model on adv. images | 0.9581 | 0.9449 | 0.9386 | 0.7435 | 0.6943 | 0.6902 | 0.662 | 0.375 | 0.206 | 0.074 |
| Robust model on substitute adv. images | 0.9665 | 0.9575 | 0.9417 | 0.7661 | 0.681 | 0.5037 | 0.767 | 0.700 | 0.380 | 0.279 |

**Analysis of making either the image channel or the edge channel in models (i.e., making them zero)**. Over the original edge augmented model (Img+Edge), the image channel is more important since masking it hurts the model more (compared to the making the edge channel). Conversely, over the robust and robust redetect models, masking the edge channel hurts more. This indicates that robust models rely more on shape than texture. Models used here are adversarially trained against each attack. For example, Img+Edge Robust model is trained separately for $\epsilon = 8/255$. This is the same setup as in the main text and tables.

Table 22: Masking channels over MNIST dataset

| | Img+Edge Model | Img+Edge Robust | | | Img+Edge Robust Redetect | | |
|---|---|---|---|---|---|---|---|
| $\epsilon$ | 0 | 8 | 32 | 64 | 8 | 32 | 64 |
| **FGSM** | | | | | | | |
| Masking Img Channels | 0.851 | 0.841 | 0.914 | 0.924 | 0.850 | 0.931 | 0.951 |
| Masking Edge Channel | 0.968 | 0.974 | 0.975 | 0.969 | 0.964 | 0.942 | 0.718 |
| **PGD** | | | | | | | |
| Masking Img Channels | 0.851 | 0.915 | 0.921 | 0.920 | 0.862 | 0.956 | 0.956 |
| Masking Edge Channel | 0.968 | 0.975 | 0.973 | 0.954 | 0.970 | 0.957 | 0.858 |

Table 23: Masking channels over Fashion MNIST dataset

| | Img+Edge Model | Img+Edge Robust | | | Img+Edge Robust Redetect | | |
|---|---|---|---|---|---|---|---|
| $\epsilon$ | 0 | 8 | 32 | 64 | 8 | 32 | 64 |
| **FGSM** | | | | | | | |
| Masking Img Channels | 0.161 | 0.324 | 0.526 | 0.690 | 0.253 | 0.560 | 0.734 |
| Masking Edge Channel | 0.768 | 0.761 | 0.709 | 0.530 | 0.715 | 0.650 | 0.543 |
| **PGD** | | | | | | | |
| Masking Img Channels | 0.161 | 0.220 | 0.585 | 0.744 | 0.246 | 0.677 | 0.760 |
| Masking Edge Channel | 0.768 | 0.758 | 0.656 | 0.100 | 0.717 | 0.577 | 0.447 |

## F   SAMPLE GENERATED IMAGES BY THE CONDITIONAL GAN IN GAN-BASED SHAPE DEFENSE (GSD)

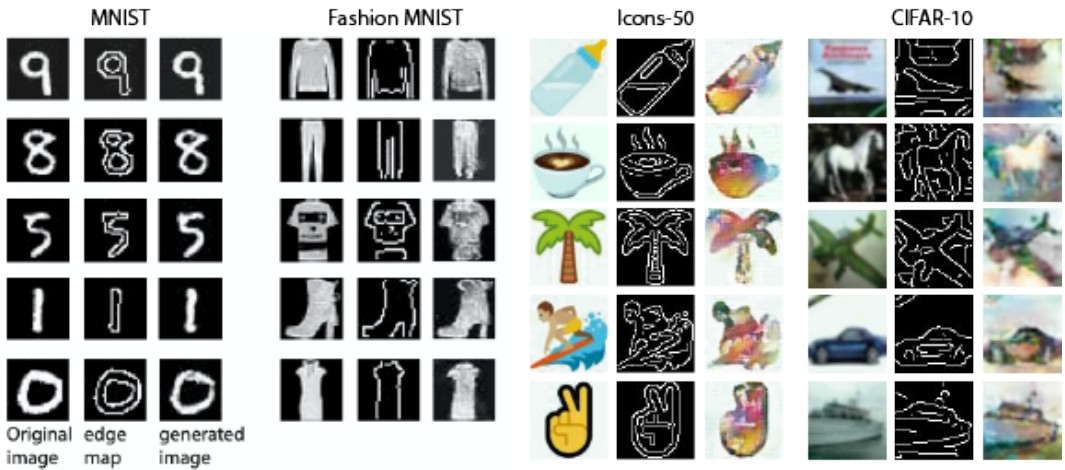

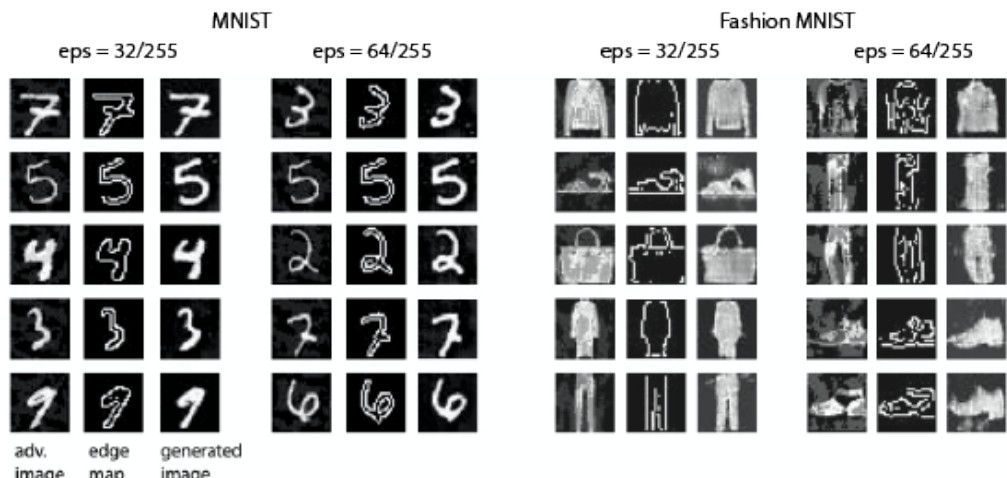

Figure 18: Top) GSD with a classifier trained on images generated (by pix2pix) only from the edge maps of the clean images, Bottom) GSD with edge maps derived from adversarial examples. Columns from left to right: adversarial images by the FGSM attack, their edge maps, and generated images by pix2pix.

## G  SHAPE-BASED EXTENSIONS OF VANILLA PGD ADVERSARIAL TRAINING, FREE ADVERSARIAL TRAINING (FREEAT), AND FAST ADVERSARIAL TRAINING (FASTAT) ALGORITHMS

---

**Algorithm 3** Shape-based PGD adversarial training for $T$ epochs, given some radius $\epsilon$, adversarial step size $\alpha$ and $N$ PGD steps and a dataset of size $M$ for a network $f_\theta$. $\beta \in \{edge, img, imgedge\}$ indicates the net_type and *redetect_train* mean edge redetection during training.

---

  **for** $t = 1 \ldots T$ **do**
    **for** $i = 1 \ldots M$ **do**
      *// Perform PGD adversarial attack*
      $\delta = 0$ *// or randomly initialized*
      **for** $j = 1 \ldots N$ **do**
        $\delta = \delta + \alpha \cdot \text{sign}(\nabla_\delta \ell(f_\theta(x_i + \delta), y_i))$
        $\delta = \max(\min(\delta, \epsilon), -\epsilon)$
      **end for**
      $\tilde{x}_i = x_i + \delta$
      **if** *redetect_train* & $\beta$ == *imgedge* **then**
        $\tilde{x}_i = \text{detect\_edge}(\tilde{x}_i)$    *// recompute and replace the edge map*
      **end if**
      $\theta = \theta - \nabla_\theta \ell(f_\theta(\tilde{x}_i), y_i)$ *// Update model weights with some optimizer, e.g. SGD*
    **end for**
  **end for**

---

**Algorithm 4** Shape-based "Free" adversarial training for $T$ epochs, given some radius $\epsilon$, $N$ minibatch replays, and a dataset of size $M$ for a network $f_\theta$. $\beta \in \{edge, img, imgedge\}$ indicates the net_type and *redetect_train* mean edge redetection during training.

---

  $\delta = 0$
  *// Iterate T/N times to account for minibatch replays and run for T total epochs*
  **for** $t = 1 \ldots T/N$ **do**
    **for** $i = 1 \ldots M$ **do**
      *// Perform simultaneous FGSM adversarial attack and model weight updates T times*
      **for** $j = 1 \ldots N$ **do**
        $\tilde{x}_i = x_i + \delta$
        **if** *redetect_train* & $\beta$ == *imgedge* **then**
          $\tilde{x}_i = \text{detect\_edge}(\tilde{x}_i)$    *// recompute and replace the edge map*
        **end if**
        *// Compute gradients for perturbation and model weights simultaneously*
        $\nabla_\delta, \nabla_\theta = \nabla \ell(f_\theta(\tilde{x}_i), y_i)$
        $\delta = \delta + \epsilon \cdot \text{sign}(\nabla_\delta)$
        $\delta = \max(\min(\delta, \epsilon), -\epsilon)$
        $\theta = \theta - \nabla_\theta$ *// Update model weights with some optimizer, e.g. SGD*
      **end for**
    **end for**
  **end for**

---

**Algorithm 5** Shape-based FGSM adversarial training for $T$ epochs, given some radius $\epsilon$, $N$ PGD steps, step size $\alpha$, and a dataset of size $M$ for a network $f_\theta$. $\beta \in \{edge, img, imgedge\}$ indicates the net_type and *redetect_train* mean edge redetection during training.

---

**for** $t = 1 \ldots T$ **do**
   **for** $i = 1 \ldots M$ **do**
      *// Perform FGSM adversarial attack*
      $\delta = \text{Uniform}(-\epsilon, \epsilon)$
      $\delta = \delta + \alpha \cdot \text{sign}(\nabla_\delta \ell(f_\theta(x_i + \delta), y_i))$
      $\delta = \max(\min(\delta, \epsilon), -\epsilon)$
      $\tilde{x}_i = x_i + \delta$
      **if** *redetect_train* & $\beta ==$ *imgedge* **then**
         $\tilde{x}_i = \text{detect\_edge}(\tilde{x}_i)$    *// recompute and replace the edge map*
      **end if**
      $\theta = \theta - \nabla_\theta \ell(f_\theta(\tilde{x}_i), y_i)$ *// Update model weights with some optimizer, e.g. SGD*
   **end for**
**end for**

---

Table 24: Performance of the Fast Adversarial Training (FastAT) method over three runs.

| Model | Run 1 Clean | Run 1 PGD-10 | Run 1 Clean | Run 1 PGD-10 | Run 1 Clean | Run 1 PGD-10 | Average Clean | Average PGD-10 |
|---|---|---|---|---|---|---|---|---|
| Edge | 0.559 | 0.384 | 0.581 | 0.187 | 0.608 | 0.586 | 0.582 | 0.386 |
| RGB | 0.813 | 0.368 | 0.598 | 0.205 | 0.889 | 0.569 | 0.767 | 0.381 |
| Img + Edge | 0.863 | 0.590 | 0.882 | 0.334 | 0.878 | 0.878 | **0.874** | 0.386 |
| Redetect | ,, | 0.593 | ,, | 0.341 | ,, | 0.245 | ,, | 0.393 |
| RGB + Redet. Edge | 0.892 | 0.001 | 0.817 | 0.115 | 0.889 | 0.105 | 0.866 | 0.074 |
| Redetect | ,, | 0.265 | ,, | 0.656 | ,, | 0.326 | ,, | **0.416** |

Table 25: Performance of the Free Adversarial Training (FreeAT) method over three runs.

| Model | Run 1 Clean | Run 1 PGD-10 | Run 1 Clean | Run 1 PGD-10 | Run 1 Clean | Run 1 PGD-10 | Average Clean | Average PGD-10 |
|---|---|---|---|---|---|---|---|---|
| Edge | 0.674 | 0.672 | 0.704 | 0.702 | 0.660 | 0.659 | 0.679 | **0.678** |
| RGB | 0.783 | 0.450 | 0.768 | 0.450 | 0.772 | 0.447 | 0.774 | 0.449 |
| Img + Edge | 0.784 | 0.432 | 0.779 | 0.447 | 0.782 | 0.448 | **0.782** | 0.442 |
| Redetect | ,, | 0.447 | ,, | 0.448 | ,, | 0.449 | ,, | 0.448 |
| RGB + Redet. Edge | 0.776 | 0.451 | 0.776 | 0.454 | 0.780 | 0.447 | 0.777 | 0.451 |
| Redetect | ,, | 0.452 | ,, | 0.456 | ,, | 0.448 | ,, | 0.452 |

# H PERFORMANCE OF THE MODELS AGAINST COMMON IMAGE CORRUPTIONS

Figure 19: Sample images alongside their corruptions with 5 severity levels.

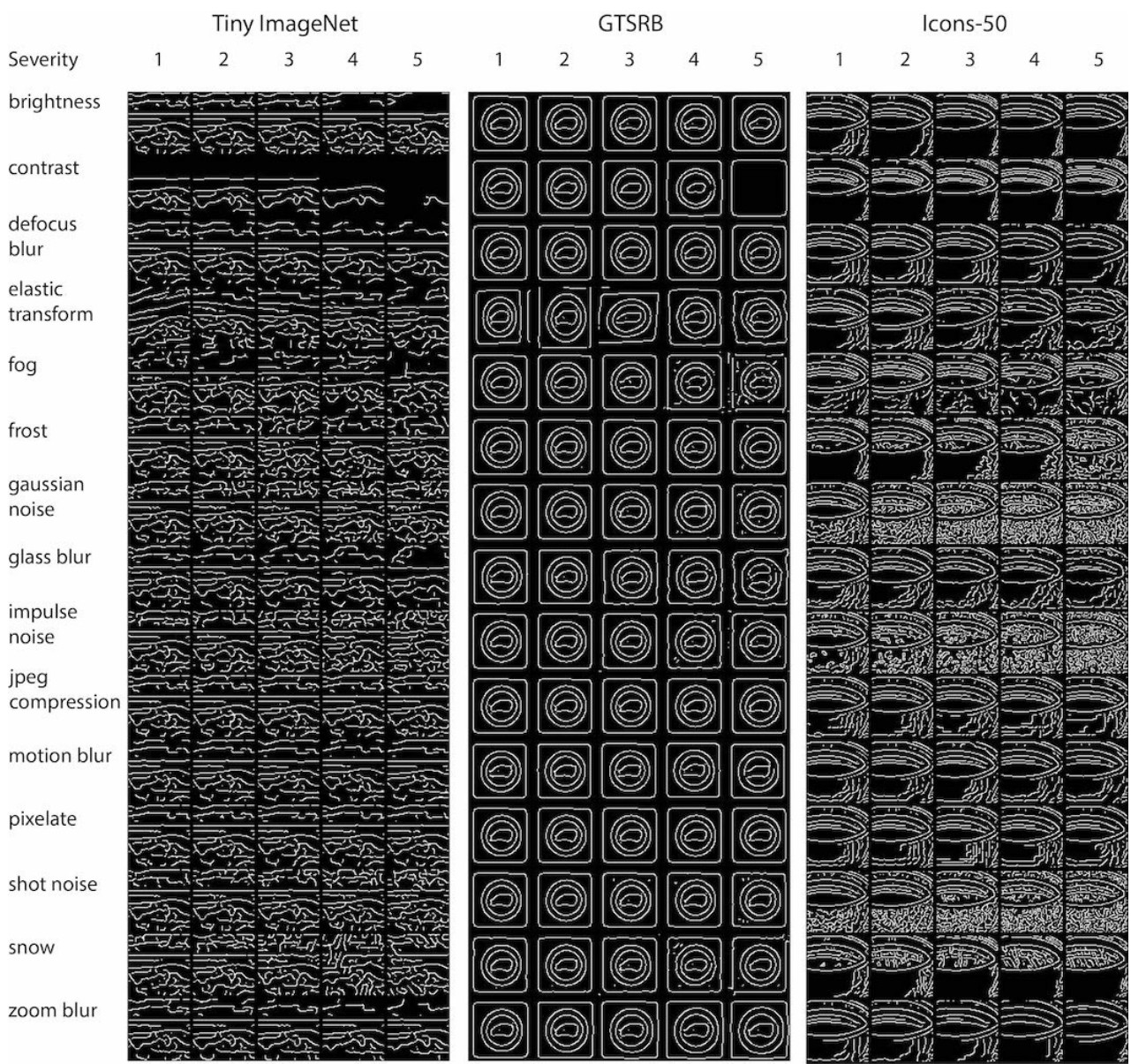

Figure 20: Edges for images in Fig. 19.

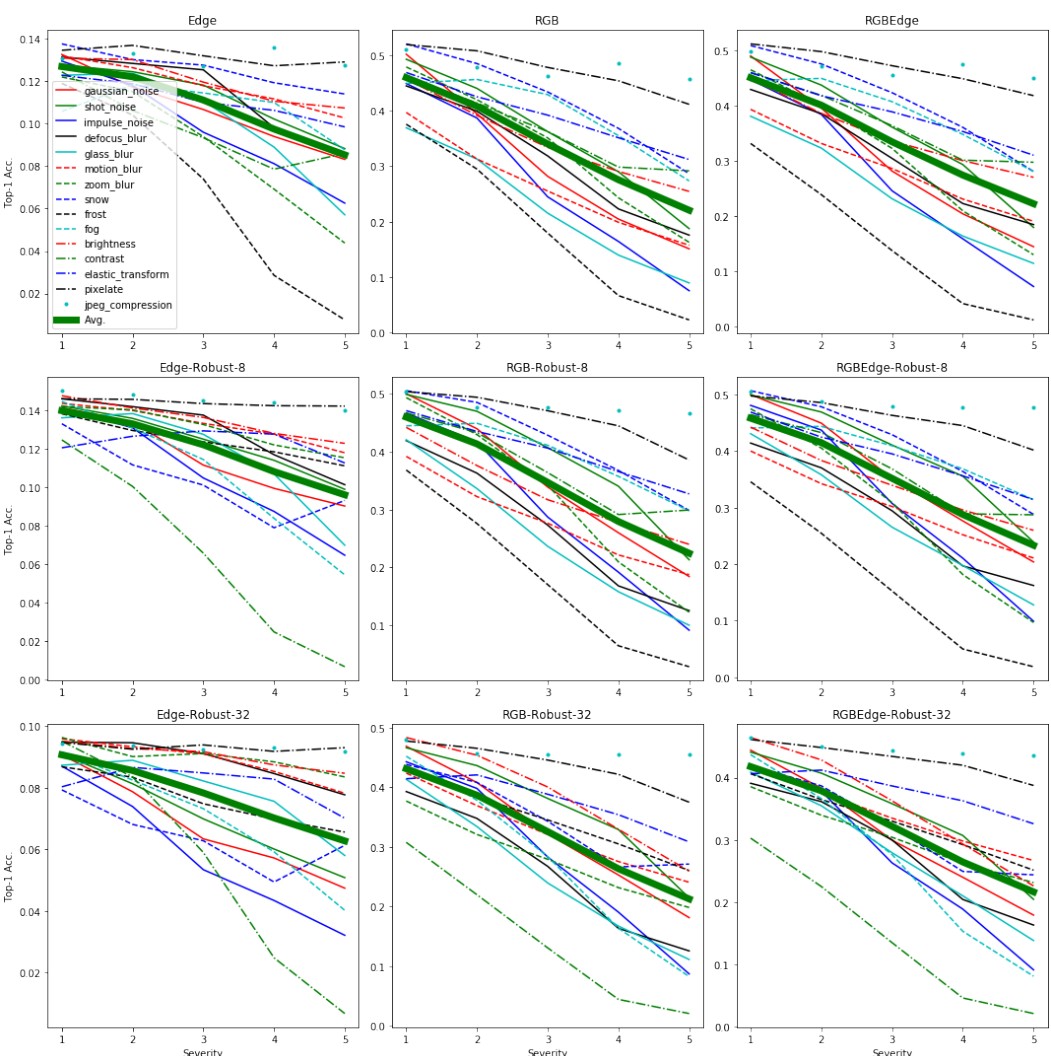

Figure 21: Performance of models against natural image corruptions over the TinyImageNet dataset. Robust models are trained against FGSM attack.

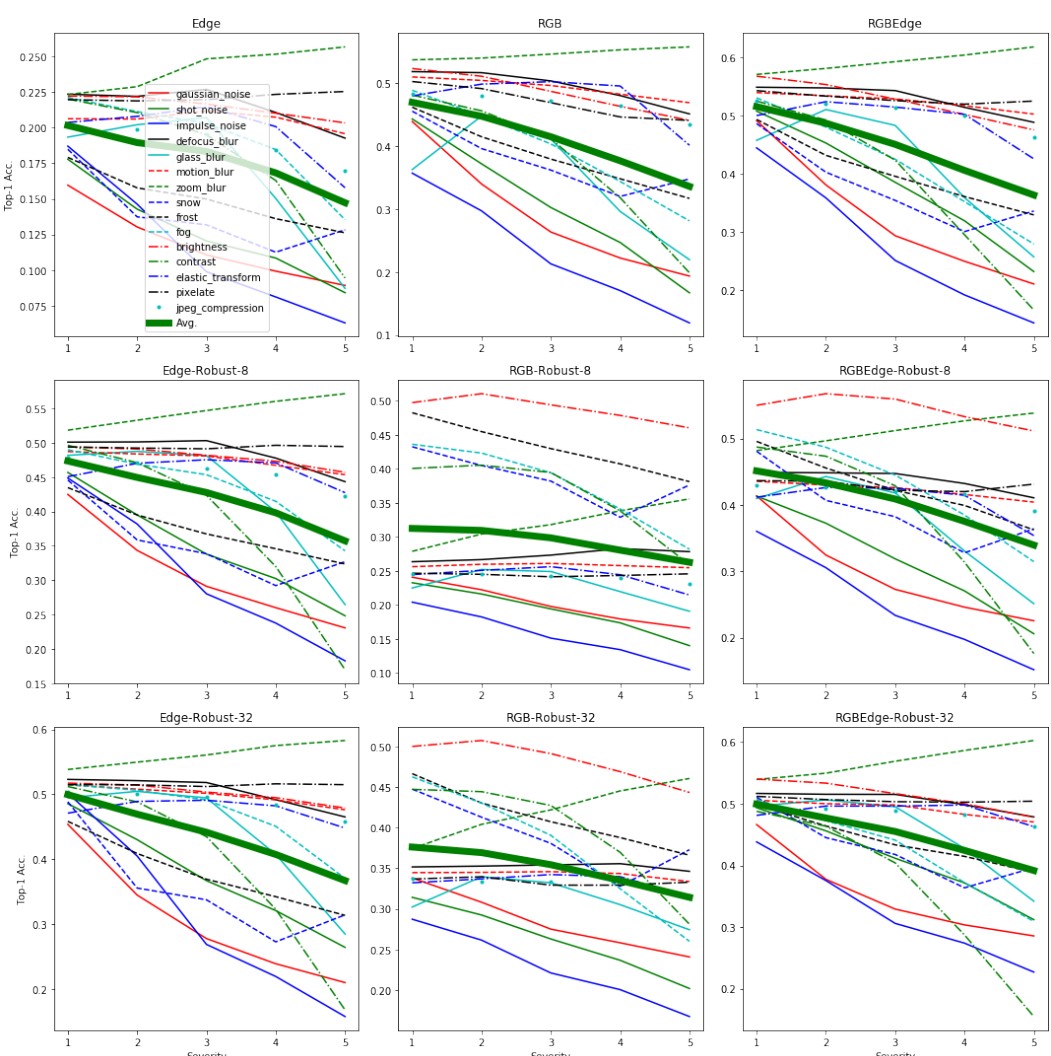

Figure 22: Performance of models against natural image corruptions over the GTSRB dataset.

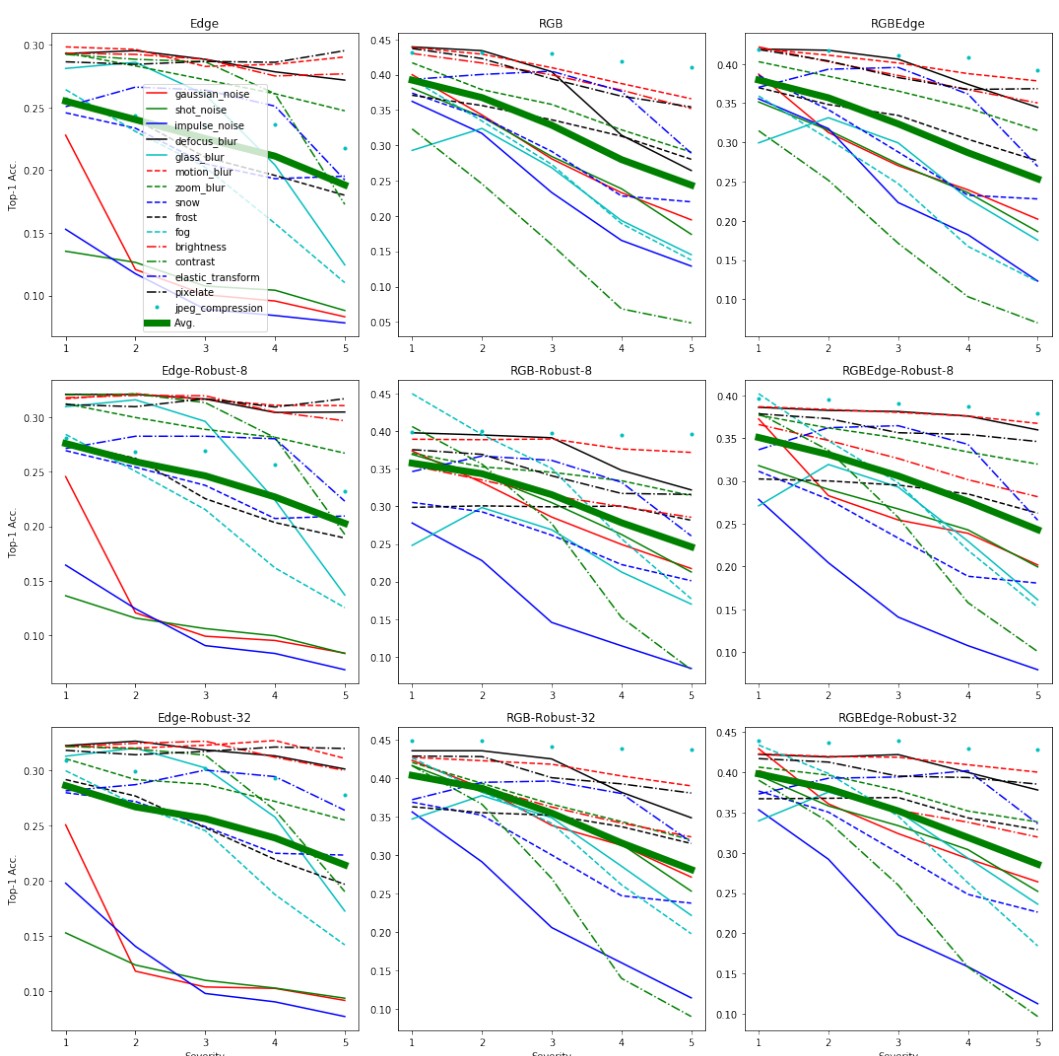

Figure 23: Performance of models against natural image corruptions over the Icons-50 dataset.

## I    EFFECT OF BACKGROUND SUBTRACTION IN ADVERSARIAL DEFENSE

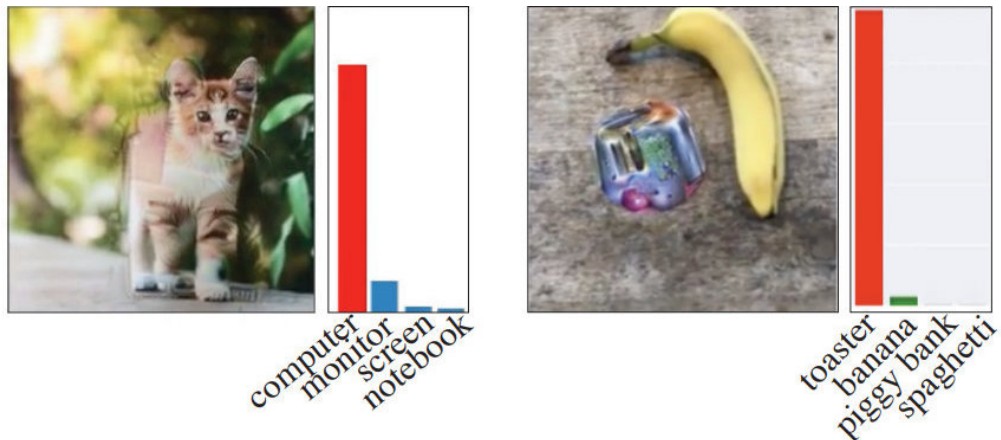

Figure 24: Examples of invisible (left) and visible (right) backdoor attacks (e.g., Brown et al. (2017)). Our proposed shape defense can easily bypass the invisible backdoor attack since edge detection removes the watermark. Our defense together with background subtraction (akin to gazing to a single object at a time) can also avoid visible backdoor attacks. See also the main text.

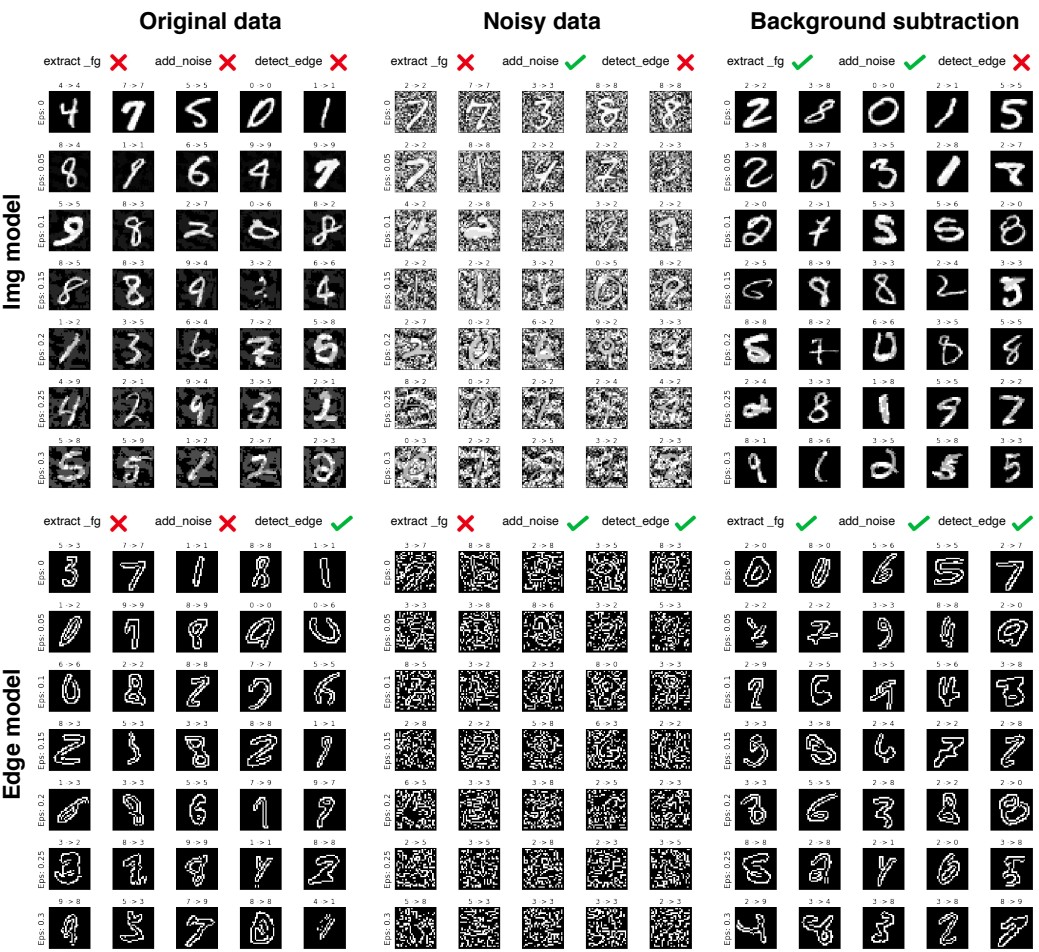

Figure 25: Application of the Img and Edge models over Original, noisy, and background-subtracted MNIST digits. Noisy data is created by overlaying a digit over white noise (noise×(1-mask)+digit). The FGSM attack is used here. We find that background subtraction together with edge detection improves robustness.

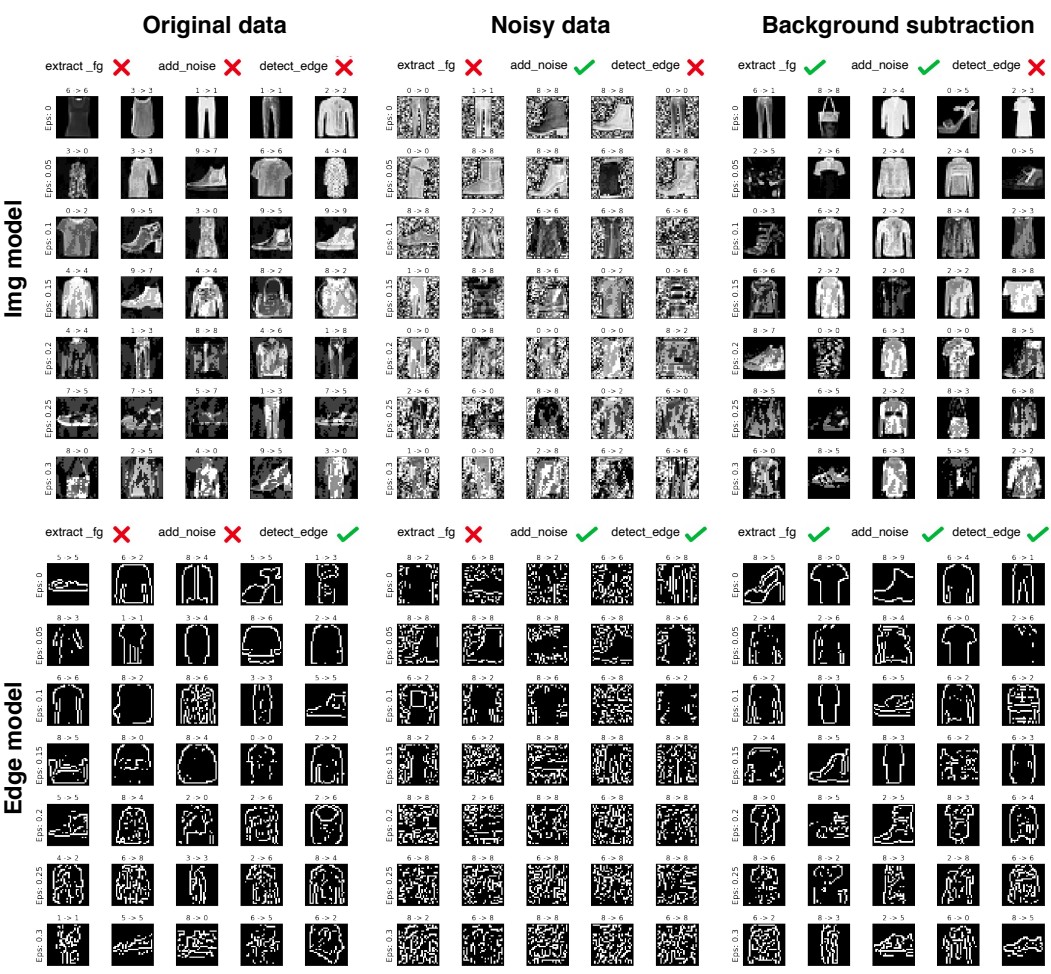

Figure 26: Application of the Img and Edge models over Original, noisy, and background-subtracted FashionMNIST data. Noisy data is created by overlaying an object over white noise (noise×(1-mask)+object). The FGSM attack is used here. We find that background subtraction together with edge detection improves robustness.

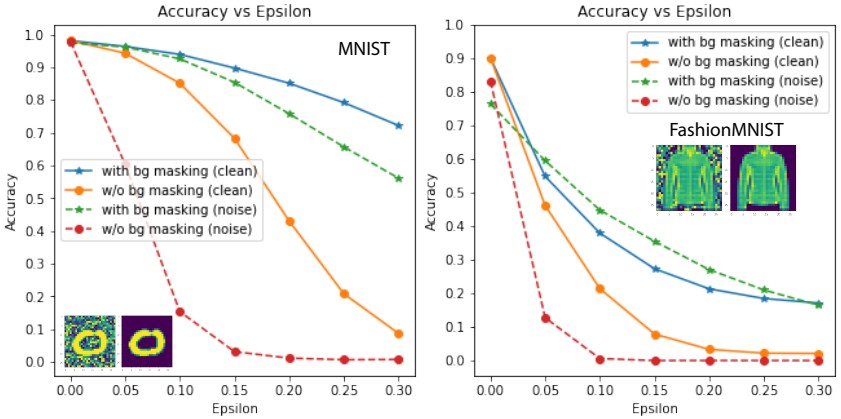

Figure 27: Similar to Fig. 5 in the main text with the difference that here the noise model is trained over the noisy data. Removing the perturbations on the image background (via background subtraction) improves the robustness.

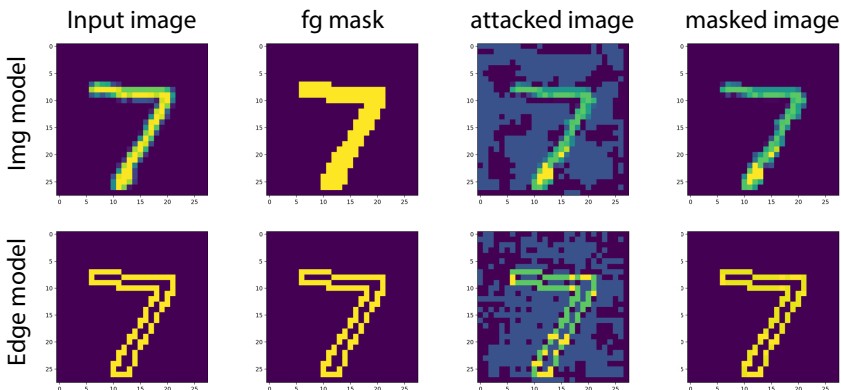

Figure 28: Masking the FGSM attack perturbations (i.e., keeping the altered pixels on the image or edge only regions).

Table 26: Performance of the models (naturally trained and adversarially-trained) against the images with only the foreground being impacted/perturbed. Compared with the results in Table. 1, applying the models to the foreground regions improves the accuracy by a large margin.

| | **FGSM** | | | **PGD-40** | | |
|---|---|---|---|---|---|---|
| | $\epsilon = 8$ | $\epsilon = 32$ | $\epsilon = 64$ | $\epsilon = 8$ | $\epsilon = 32$ | $\epsilon = 64$ |
| **Img (natural training)** | 0.9598 | 0.9021 | 0.7516 | 0.9598 | 0.8947 | 0.7042 |
| **Img (adversarial training)** | 0.9641 | 0.9237 | 0.8398 | 0.9636 | 0.9139 | 0.7868 |
| **Edge (natural training)** | 0.9553 | 0.9199 | 0.8323 | 0.9553 | 0.9213 | 0.8452 |
| **Edge (adversarial training)** | 0.9686 | 0.9468 | 0.9059 | 0.9686 | 0.9474 | 0.9057 |

## J    ROBUSTNESS AGAINST THE CW ATTACK OVER MNIST DATASET

Performance of the the EAT defense against the $l_2$ Carlini-Wagner attack (Carlini & Wagner, 2017) with the following parameters:

```
attack = CW(net, targeted=False, c=1e-4, kappa=0, iters=10, lr=0.001)
```

| | Orig. model | | Robust model | | Average |
| --- | --- | --- | --- | --- | --- |
| | 0/clean | adv. | 0/clean | adv. | Rob. models |
| **Edge** | 0.964 | 0.106 | 0.948 | 0.798 | 0.873 |
| **Img2Edge** | ,, | 0.962 | ,, | 0.949 | **0.949** |
| **Img** | 0.973 | 0.103 | 0.949 | 0.856 | 0.903 |
| **Img+Edge** | 0.972 | 0.097 | 0.945 | 0.845 | 0.895 |
| Redetect | ,, | 0.971 | ,, | 0.942 | 0.944 |
| **Img + Redetected Edge** | | | 0.947 | 0.819 | 0.883 |
| Redetect | | | ,, | 0.946 | 0.946 |

## K  ROBUSTNESS AGAINST BOUNDARY ATTACK

Performance of the the edge augmented model against the Boundary attack (Brendel et al., 2017) with the following parameters:

```
BoundaryAttack(init_attack=None, steps=25000, spherical_step=0.01,
               source_step=0.01, source_step_convergance=1e-07,
               step_adaptation=1.5, tensorboard=False,
               update_stats_every_k=10)
```

Table 27: Results over 500 images from the MNIST dataset

|  | Orig. model | |
| --- | --- | --- |
|  | 0/clean | adv. (boundary) |
| **Edge** | 0.964 | 0.000 |
| **Img** | 0.973 | 0.003 |
| **Img+Edge** | 0.972 | 0.000 |
| Redetect | ,, | **0.945** |
| **Img+Redetected Edge (adversarially trained using FGSM $\epsilon = 8/255$)** | 0.974 | 0.001 |
| Redetect | ,, | **0.965** |

Table 28: Results over 500 images from the Fashion MNIST dataset

|  | Orig. model | |
| --- | --- | --- |
|  | 0/clean | adv. (boundary) |
| **Edge** | 0.776 | 0.005 |
| **Img** | 0.798 | 0.018 |
| **Img+Edge** | 0.809 | 0.003 |
| Redetect | ,, | **0.747** |
| **Img+Redetected Edge (adversarially trained using FGSM $\epsilon = 8/255$)** | 0.789 | 0.003 |
| Redetect | ,, | **0.770** |

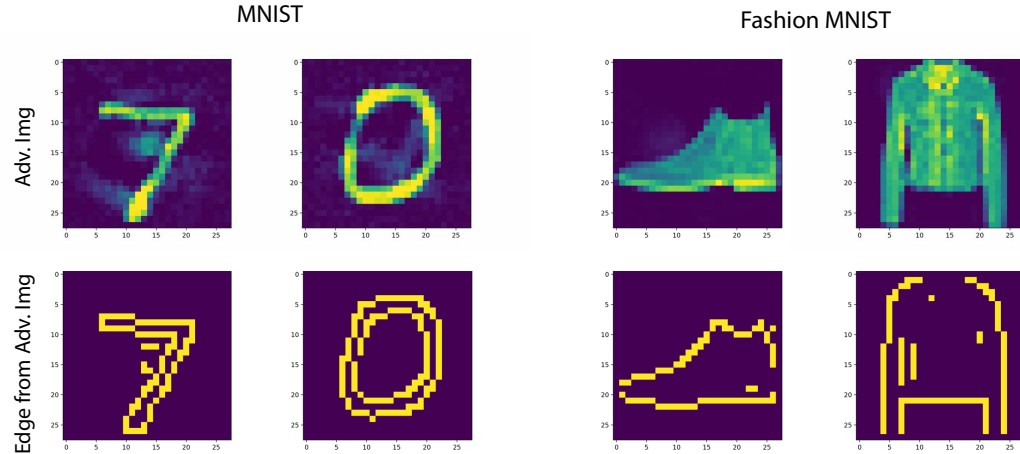

Figure 29: Sample images from the Boundary attack.

## L  ROBUSTNESS AGAINST ADAPTIVE ATTACKS OVER IMAGENETTE2-160 DATASET

We use the PyTorch implementation[6] of the HED edge detector proposed by Xie & Tu (2015). Here, a classifier is first trained on top of the edge maps from the HED. Then, the entire pipeline (Img $\longrightarrow$ HED $\longrightarrow$ Classifier$^{HED}$) is attacked to generate an adversarial image. The performance of this classifier is measured on both clean and adversarial images. The adversarial image is also fed to the classifier trained on Canny edge maps (Img$^{adv-HED}$ $\longrightarrow$ Canny $\longrightarrow$ Classifier$^{Canny}$). Results are shown in Table below. As it can be seen, adversarial examples crafted for HED fail to completely fool the model trained on Canny edges (i.e., they do not transfer).

Table 29: Results over 500 images from the Imagenette2-160 dataset against the FGSM and PGD-5 ($\epsilon = 8/255$) attacks.

|  | Orig. model | | |
| --- | --- | --- | --- |
|  | 0/clean | adv. (FGSM) | adv. (PGD-5) |
| **Img2Edge (Img $\longrightarrow$ HED $\longrightarrow$ Classifier$^{HED}$)** | 0.793 | 0.052 | 0.003 |
| **Img2Edge (Img$^{adv-HED}$ $\longrightarrow$ Canny $\longrightarrow$ Classifier$^{Canny}$)** | 0.767 | 0.542 | 0.548 |

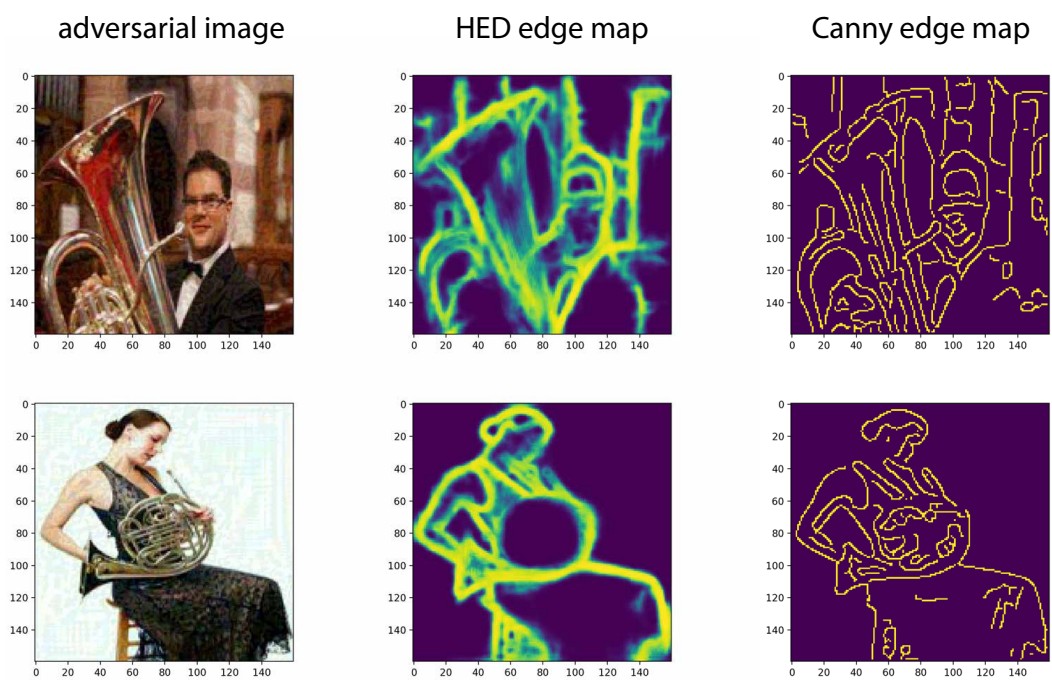

Figure 30: Two sample adversarial images (FGSM) along with their edge maps using HED and Canny edge detection methods.

---

[6]https://github.com/sniklaus/pytorch-hed

## M    ROBUSTNESS AGAINST ADAPTIVE ATTACKS OVER MNIST DATASET

Here, we attempt to explicitly approximate the Canny edge detector using a differentiable convolutional autoencoder. In our pipeline, a classifier (CNN) is stacked after the convolutional autoencoder (with sigmoid output neurons). We first freeze the classifier and train the autoencoder using the MSE loss with (input, output) pair being (image, canny edge map). We then freeze the autoencoder and train the classifier using Cross Entropy loss. After training the network, we then craft adversarial examples for it and feed them to a classifier trained on Canny edges (original models or robust models as was mentioned in the main text). Fig. 31 shows the pipeline and some sample approximated edge maps. Fig. 32 shows the architecture details in PyTorch.

The top panel in Fig. 33 shows results using the FGSM and PGD-40 attacks against the pipeline itself, and also against the Img2Edge model (trained over clean edges or adversarial ones[7]). As can be seen, both attacks are very successful against the pipeline but they do not perform well against the Canny edge map classifier (i.e., crafted adversarial examples for the pipeline do not transfer well to the Imge2Edge trained over Canny Edge map; img$\longrightarrow$ Canny $\longrightarrow$ class label). Notice, that here we only used the model trained on edge maps. It is likely to gain even better robustness against the adaptive attacks in using the img+edge+redetect.

The bottom panel in Fig. 33 shows sample adversarial digits (constructed using the adaptive attack) and their edge maps under the FGSM and PGD-40 attacks. Notice how PGD-40 attack preserves the edges (compered to FGSM). This is because it needs less perturbation to fool the classifier. Also, notice that the perturbations shown are perceptible which results in edges maps having noise. If we limit ourselves to imperceptible perturbations, then edge maps will not change much compared to the original edge maps on clean images.

---

[7]Here we used the model adversarially trained at eps=8/255 and test it against other perturbations; unlike the main text where we trained robust models separately for each epsilon.

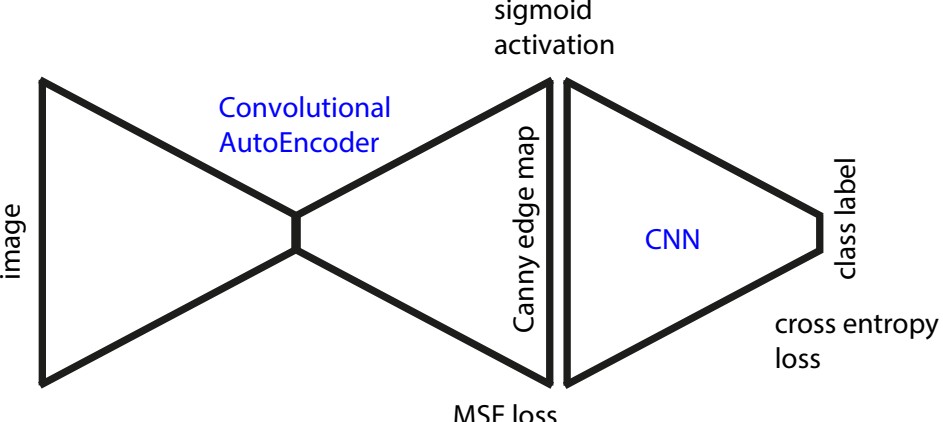

1. Freeze the CNN (requires_grad = False) and train the AutoEncoder
2. Freeze the AutoEncoder (requires_grad = False) and train the CNN
3. Unfreeze all the network (requires_grad = True) and attack it
4. Feed the adeversarial image to a CNN trained with Canny edge maps

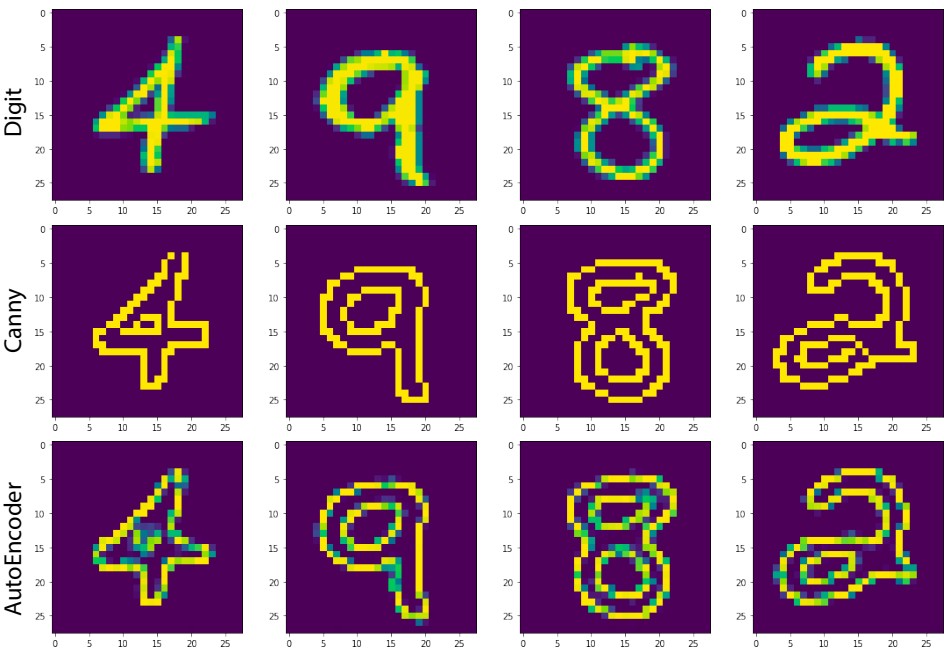

Figure 31: Top: our pipeline to approximate the Canny edge detector and our approach for crafting adversarial examples, Bottom: Sample digits and their generated edge maps.

```python
# combined network
# LeNet Model definition
class MNIST_Net_combined(nn.Module):
    def __init__(self, net_type='gray'):
        super(MNIST_Net_combined, self).__init__()

        self.encoder = nn.Sequential( # like the Composition layer you built
            nn.Conv2d(1, 16, 3, stride=2, padding=1),
            nn.ReLU(),
            nn.Conv2d(16, 32, 3, stride=2, padding=1),
            nn.ReLU(),
            nn.Conv2d(32, 64, 7)
        )
        self.decoder = nn.Sequential(
            nn.ConvTranspose2d(64, 32, 7),
            nn.ReLU(),
            nn.ConvTranspose2d(32, 16, 3, stride=2, padding=1, output_padding=1),
            nn.ReLU(),
            nn.ConvTranspose2d(16, 1, 3, stride=2, padding=1, output_padding=1),
            nn.Sigmoid()
        )

        self.conv1 = nn.Conv2d(1, 10, kernel_size=5)
        self.conv2 = nn.Conv2d(10, 20, kernel_size=5)
        self.conv2_drop = nn.Dropout2d()
        self.fc1 = nn.Linear(320, 50)
        self.fc2 = nn.Linear(50, 10)

    def forward(self, x):
        z = self.encoder(x)
        x_auto = self.decoder(z) # reconstructed egde
        x_auto = x_auto.view(x_auto.shape[0],1, 28,28)
        x = F.relu(F.max_pool2d(self.conv1(x_auto), 2))
        x = F.relu(F.max_pool2d(self.conv2_drop(self.conv2(x)), 2))
        x = x.view(-1, 320)
        x = F.relu(self.fc1(x))
        x = F.dropout(x, training=self.training)
        x = self.fc2(x)
        return x, x_auto
```

Figure 32: PyTorch code of our pipeline shown in Fig 31.

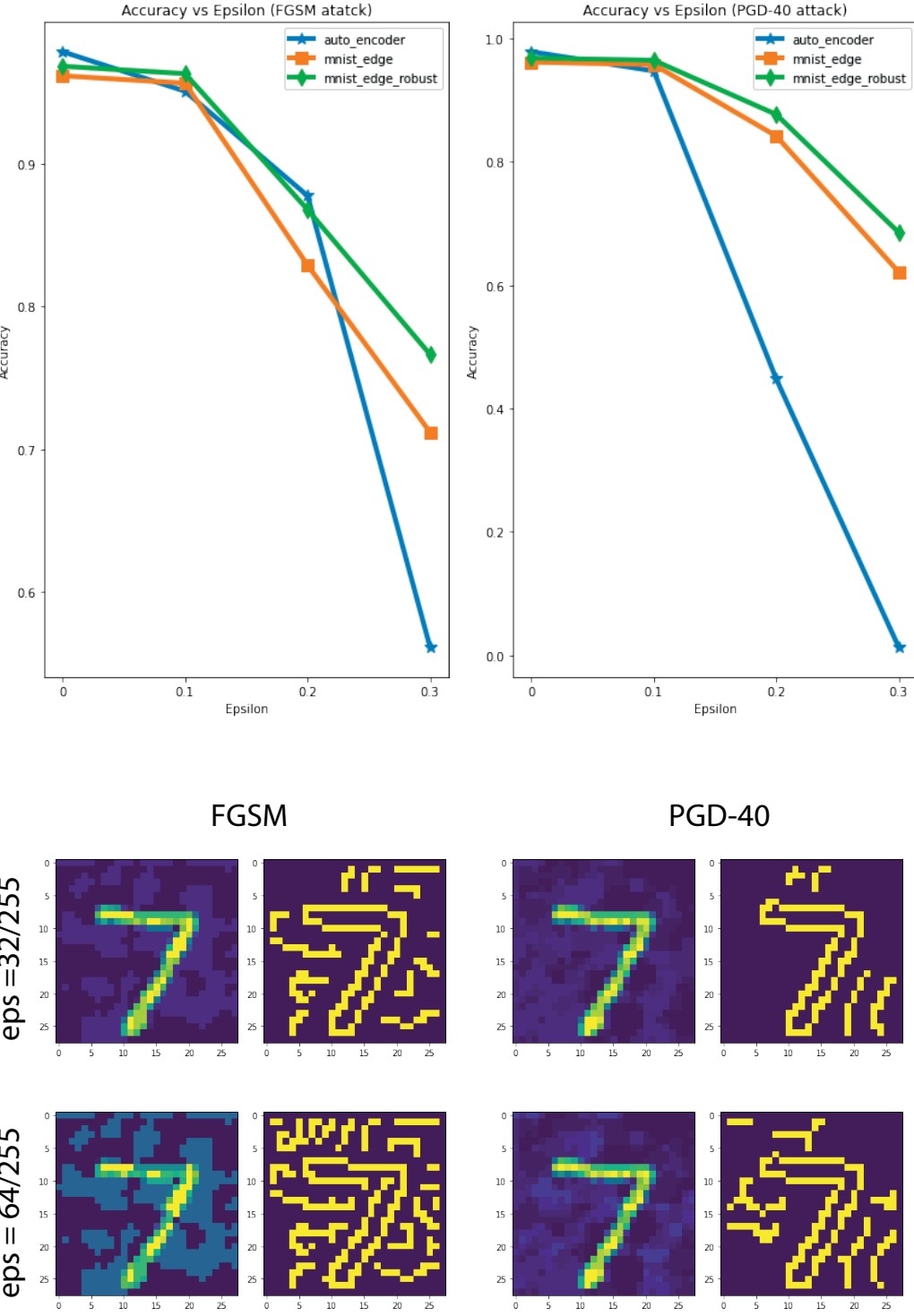

Figure 33: Top: Performance of the adaptive attack, Bottom: Samples adversarial images and their edge maps using the Canny edge detector.

