# OpenReview forum: "Shape Defense"
_ICLR.cc/2021/Conference — Reject_

### Official Review · AnonReviewer3 · 2020-10-28
**Repeated work and limited contributions.**

**Rating:** 3
**Confidence:** 4

**Review:**

This paper investigates incorporating shape information in deep neural networks to improve their adversarial robustness. It proposes two methods: the first one is to augment the input with the corresponding edge and then adversarially train a CNN on the augmented input. The second idea is to train a conditional GAN to reconstruct images from edge maps and use the reconstructed image as input to a standard classifier.

1. The description of the proposed defense in section 3 seems to be limited. It is not clear why the author applied a conditional GAN to reconstruct clean images from edge maps. In other words, what is the motivation for designing GSD on top of EAT?

2. The authors use Canny edge detector to extract edges. Why not use neural network based edge extractors [2] as they give better edges? What is the motivation here?

3. Considering the possible obfuscated gradient issues of white-box attacks [3], the authors should explicitly describe their efforts to evaluate against strong custom adaptive attacks.

4. In terms of the experiments, the authors claim that they investigated adaptive attack but I did not see any quantitative experiment results. They also claim that any adaptive attack would cause perceptible changes to the edges. This is not an excuse for not doing quantitative study; the authors already considered adversarial perturbations with magnitude as large as 64. Such magnitude can also cause perceptible changes to images as Figure 8 shows.

5. For EAT, what is the performance if the model is not adversarially trained? Why use adversarial training in EAT but not in GSD? I believe these analyses are required for an in-depth understanding of how the proposed defense works.

6. Last but not least, the algorithms proposed in this paper looks similar (almost the same) to this paper [1] from previous year: (a) The edge-guided adversarial training (EST) is basically applying adversarial training on EdgeNetRob in [1]; (b) The GAN-based shape defense (GSD) is exactly the same as EdgeGANRob in [1]; (c) Both of them use canny edge detector to extract edges. Can the authors highlight the differences? If this is a separate paper, given the previous work [1] that already proposed this idea, the contribution of this work seems to be limited.

[1] Shape Features Improve General Model Robustness. https://openreview.net/forum?id=SJlPZlStwS, 2019.

[2] Richer Convolutional Features for Edge Detection. Liu, et al TPAMI, 2019.

[3] Obfuscated Gradients Give a False Sense of Security: Circumventing Defenses to Adversarial Examples. Anish et al, ICML 2018.

---

> ### Author Response · Authors · 2020-11-13
> **Adaptive attacks and similar work**
>
> Thanks for your review and your detailed comments.
>
> 1. GSD is not on top of EAT and does not extend it. As is mentioned, our goal is to purify the input using GSD. We were inspired by the pix2pix paper where edges were converted to the rgb images. So if the edge map stays the same for an imperceptible perturbation it should result it in a good generated image and hence accurate classification by a classifier trained on generated images and/or original images. We will expand this section.
>
>
> 2. Proposed methods can be used with any edge detection model. The reason we choose Canny is because it is easy to use and it is hard to attack it since it has a complicated algorithm. Nonetheless, we are going to use deep learning edge detectors and attack the entire pipeline. We will also include results against other strong attacks such as the boundary attack.
>
>
> 3 & 4. As an additional analysis against adaptive attacks, we will use deep edge detectors and perform attacks against the entire pipeline. We will also consider other attacks such as the boundary attacks and also gradient free attacks. See rev #3's comments.
>
> 5. Results of EAT over original images (without adversarial training) is shown on the left columns for all datasets. We do perform adversarial training over GSD. Please see Fig. 4. Stars (over MNIST and FashionMNIST) show performance when a classifier is trained over images generated from edge maps of original and perturbed images. As it can be seen adversarial training of GSD improves the robustness.
>
>
> 6. Thanks for the pointer. Indeed this work is similar to ours but we have invented the proposed algorithms independently and were not aware of this work at the time of the submission. We went through this paper. First, it has been withdrawn from the last year ICLR and seems not to be published elsewhere. Therefore, it is not possible to assess their methods. Nonetheless, there are significant differences between our methods and theirs as explained below:
> - We concat the edge map in depth to the rgb image whereas they train a classifier on the edge map. Their approach is just one of our baselines.
> - Our approach (EAT) allows adversarial training whereas EdgeNetRob is just a classifier on top of the edge map.
> - They propose a customized canny edge detector to make it robust, whereas we use the basic Canny or Sobel edge detectors.
> - We evaluate our approach against common natural perturbations
> - We motivate the observation that edge map is invariant to adversarial perturbations
> - We extend other defenses such as FAST adversarial training and Free adversarial training
> - They present results over Fashion MNIST and CIFAR10 whereas here we present results over 10 datasets (including natural scenes)
> - We propose foreground detection to defend against backdoor attacks in conjunction with edge detection
> - Compared to their GAN based method, we use natural scenes whereas they use Fashion MNIST and Celeb dataset which are much easier than CIFAR and GTSRB
> - We evaluate models over a wide range of parameters and attacks
>
> We will certainly also discuss the mentioned paper in the discussion section and compare our work with theirs.

---

> > ### Author Response · Authors · 2020-11-18
> > **Adaptive attacks and using deep edge detectors**
> >
> > According to your suggestion we now use a deep edge detector (holistically-nested edge detector; HED) to devise an adaptive attack. Please see Appendix L in the paper. In short, we used the HED edge detector and built a classifier on it. We then fed the generated adversarial images to the original model (based on the Canny edge detector). Even without adversarial training, the original model is robust to such adversarial examples (some are given in Fig. 30 in Appx L). We expect better results using adversarial training. The results shown in row 2 in Table 29 are about the same as when using the original model shown in Table 17 which shows that attack is effective around 60% of the time on the used dataset (so adaptive attack has not been successful). We believe the reason why the proposed defense works against such an adaptive attack is because it generates a binary edge map which retains the most important (and perhaps minimal information) from the image to classify the input. Any adaptive attacks has to change the image such that its Canny edge map is significantly changed (notice again that the edge map is binary not real-valued).
> >
> > We have also adequately discussed and referenced the mentioned study in related works.

---

> > > ### Comment · AnonReviewer3 · 2020-11-21
> > > **Concerns about experiment settings**
> > >
> > > I would like to thank the authors for conducting additional studies and making clarifications. However, I have concerns about the experiment settings of the adaptive attack and claims made by the authors based on the experiments:
> > > 1. The authors claim that they did adaptive attack by approximating the Canny edge detector with HED. However, HED is not a good approximation of Canny edge detector. The ways Canny edge detector and HED work are worlds apart and this is also reflected in the drastic visual appearance difference between the edge maps produced by Canny edge detector and HED as Figure 30 shows.
> > > 2. Besides, I’m confused by the experiment setting of Img^{adv-HED} ->Canny->Classifier^{Canny}. Why do the aurthors want to feed the adversarial images generated by one model, Classifier^{HED}, to a different model Classifer^{Canny}? The experiment results can only show the weak transferability of the adversarial perturbations across models.
> > > 3. Moreover, the authors repeatedly claim that the proposed defense works because it is biased toward shape and agrees with human vision. From a human perception perspective, the HED edge map of the adversarial image is as valid a shape/contour representation of the image as the Canny edge map in Figure 30. Then the authors need to explain why the HED edge map can still successfully attack the model as the first row of Table 29 shows.
> > > Before the concerns can be resolved I do not agree with the authors conclusion that the proposed method is robust against adaptive attack even without considering the drop of accuracy in the setting of Image^{adv-HED}->Canny->Classifier^{Canny}. In a strict sense of adaptive attack, the experiment results could imply the opposite of the authors’ conclusion, the proposed defense is not robust against adaptive attack when using a deep edge detector.
> > >
> > > I also need more clarification on the experiment setting of boundary attack. Are the edges of each image redetected during the attack? If edges are redetected during attack, why the adversarial images can successfully attack the model in the Img+Edge row but fail in the redetect row? If they are not redetected, then why? Edge detection is the part in the model that can not be differentiated through and where applying gradient-free attack becomes meaningful.

---

> > > > ### Author Response · Authors · 2020-11-23
> > > > **Your concerns on the adaptive attack**
> > > >
> > > > Thanks for your comments.
> > > >
> > > > 1. Good point! As is mentioned in page 6 of the paper, the adversary has to approximate the edge detector such that it is also differentiable. We used HED off the shelf. Retraining it on natural images to approximate Canny is a bit complicated and perhaps needs a lot of time. Instead, we have devised a second experiment in which we actually train a convolutional auto encoder to approximate Canny and then append a classifier on top of it. We then use this pipeline to generate adversarial examples to attack the original system based on Canny. Details are given in page 6 and Appx. M. As you can see, the autoencoder approximates Canny very well. According to Fig. 33, the adaptive attack does not do well. Notice that we have used the Img2Edge model here without augmentation and edge redetection. Doing so may make the model even more robust. As is discussed in the paper, robustness is gained mainly due to the binary nature of the Canny edge detector.
> > > > We will elaborate on this below.
> > > >
> > > > 2. An adaptive attack is the one specifically designed to target a given defense. Here our defense is based on Canny edge detection. That is why we are trying to approximate it by a differentiable operation. This by no means is the only way to incorporate edges into a classifier. One might even use a deep edge detector for this purpose. Based on our experiments (an idea we have not tested yet),  thresholding the output of the deep edge detector (at test time) before passing it to the edge classifier may improve robustness. In other words, the adversary can craft an adversarial example from the network but at test time the output of the edge map will be thresholded (\ie hard threshold instead of the sigmoid).
> > > >
> > > > 3. True that both HED and Canny make the models rely on shape and that both types of edge detectors align with our perception of edges. However, what is shown in Table 29 is that the pipeline on HED can be attacked since it is differentiable. HED does not work because it generates continuous-valued edge maps. In other words, the input image can be changed such that some specific values in the edge map are changed (unlike Canny). Why then Canny works? Because it generates binary edge maps which results in purifying the input image since imperceptible perturbations do not affect the edge map much. This is the main point of the paper. Now, to build a completely robust system that relies of edge, we need a robust edge detector. An important point to notice here is classifier relying on binary edge map is biased towards shape leaving less opportunity for the adversary, since adversary has to change those pixels along the edges to fool the classifier.
> > > >
> > > > 4. We do not perform edge reduction inside the Boundary attack (no redetect). We use foolbox as follows to implement this attack:
> > > >             fmodel = fb.PyTorchModel(net, bounds=(0, 1))
> > > >             attack = fb.attacks.BoundaryAttack(steps=25000)
> > > >
> > > > Implementing edge redetect inside the Boundary attack does not seem to be easy. Even if someone does so, edge redetection at each time step, will make the algorithm land in a different location in feature space. This somewhat disrupt the attack's goal which is moving along the class boundary (as is illustrated in Fig 2. Brendel et al. ICLR 2018). In other words, although the model is fixed, edge redetection confuses the attack forcing it to jump around in the feature space. Although not a direct answer, we additionally measured the performance of the Boundary attack on models that have been trained on Img + Edge redetect. Results are shown in the last rows of Tables 27 and 28. As before, we find that Boundary attack fails the model, but performance is recovered after edge redetection. Overall, our exploration shows that it is not easy to fool the model but of course there might be ways to do so by tweaking the attacks. One point that is clear to us is that shape information is an invariant dimension that must be explored to build robust models.
> > > >
> > > >
> > > >
> > > > Overall, we have tried hard to break this defense with a variety of attacks. Like other defenses, shape defense might also be vulnerable to certain types of adaptive attacks (or Boundary attack for that matter). Beyond the strong results reported here, our work has a broader message that is shape is an important cue in gaining robustness and must be utilized in adversarial defenses (since edge maps of adversarial examples produced by attacks looks the same to humans). Our work encourages future explorations in this direction.
> > > >
> > > > Please see the updated manuscript.

---

### Official Review · AnonReviewer2 · 2020-10-30
**Official Blind Review #2**

**Rating:** 4
**Confidence:** 4

**Review:**

Summary:
- In this paper, the authors try to learn robust models for visual recognition and propose two defense methods, Edge-guided Adversarial Training and GAN-based Shape Defense (GSD), to use shape bias and background subtraction to strengthen the model robustness.

However, I have still some concerns below:
- In summary, the paper is hard to follow and the writting is not clear, such as the detailed motivation of the proposed methods and the structure of this paper.
- For the experiments, a big dataset is needed, such as CIFAR-100. In addition, the results are not convincing, i.e., the evaluation on FGSM and PGD attack is not enough, some gradient-free attacks are needed.

---

> ### Author Response · Authors · 2020-11-12
> **Clarity of the paper and gradient-free attacks**
>
> Thank you for your review!
>
> Interestingly, the other reviewers find the writing and structure of the paper very clear. We motivate the idea clearly in Fig. 1.
> We will expand some parts of the paper (since an additional page will be allowed) and will move Fig. 7 in the supplement to the main text.
>
> Regarding need for big datasets: We already have 10 datasets (with different image resolutions) and some of them are bigger than CIFAR-100!
>
> Regarding gradient-free attacks: As mentioned in the paper, our main goal was to show that incorporating edges helps adversarial robustness over a range of attacks including white-box, black-box, and backdoor as well as common image corruptions. We did not aim to optimize the approach to beat all other defenses. Gradient free methods are slow and hard to conduct. Further, attacks such as the one-pixel attack, will not change the edge map. Therefore, it is very likely that they wont be effective against our proposed approaches. For instance, a classifier trained on the edge map will not be affected by only changing one pixel in the image. In other words, the edge map will be the same.
>
> Our defenses offer new ideas and are different from the existing approaches.

---

> > ### Public Comment · ~Nicholas_Carlini1 · 2020-11-12
> > **Gradient-free attacks are worthwhile**
> >
> > Gradient-free attacks are worth evaluating against. Especially for defenses that are difficult to differentiate through (the edge detector here seems to qualify), showing that the defense is robust to the attacker who tries simple methods to evade it is necessary. While the one pixel attack, as you say, might not work, there are other gradient free attacks. Does your method hold up against the Boundary Attack (or its variants, e.g., HopSkipJump) or against confidence-based attacks like SPSA?
> >
> > If trying these gradient free attacks is not possible, how does the defense hold up against an attacker who tries to construct a nicely differentiable version of the edge detector and then compute gradients through both that and the neural network?

---

> > > ### Author Response · Authors · 2020-11-18
> > > **Creating a nicely differentiable version of the edge detector and robustness against gradient-free attacks**
> > >
> > > Thanks for your comments. We investigated robustness against both the Boundary Attack (as suggested) as well as an adaptive attack. Details can be found in Appendix L in the paper. In short, we used the HED edge detector and built a classifier on it. We then fed the generated adversarial images to the original model (based on the Canny edge detector). Even without adversarial training, the original model is robust to such adversarial examples (some are given in Fig. 30 in Appx L). We expect better results using adversarial training. The results shown in row 2 in Table 29 are about the same as when using the original model shown in Table 17 which shows that attack is effective around 60% of the time on the used dataset (so adaptive attack has not been successful). We believe the reason why the proposed defense works against such an adaptive attack is because it generates a binary edge map which retains the most important (and perhaps minimal information) from the image to classify the input. Any adaptive attacks has to change the image such that its Canny edge map is significantly changed (notice again that the edge map is binary not real-valued).

---

> > > > ### Comment · AnonReviewer2 · 2020-11-25
> > > > **How about the evaluation on query-based attacks.**
> > > >
> > > > Could the authors show the evaluation results on the query-based attacks, such as Nattack[1]?
> > > >
> > > >
> > > > [1] NATTACK: Learning the Distributions of Adversarial Examples for an Improved Black-Box Attack on Deep Neural Networks. ICML 2019.

---

> > ### Author Response · Authors · 2020-11-17
> > **Robustness against gradient-free attacks and adaptive attacks**
> >
> > We have assessed the performance of the proposed approach against an effective decision-based gradient free attack known as the Boundary attack (Brendel et al., 2017). Please see page 6 and appendix K in the paper. Even without adversarial training and using the augmented image+edge model that does edge redetection at inference time fails the boundary attack. Over MNIST and Fashion MNIST datasets, the model still performs very well. Inspecting the images attacked by the boundary attack (Fig. 29 in Appx. K) , shows that Canny edge map remains the same after the attack which is why model can survive the attack. Notice that perturbation by the Boundary attack is actually not imperceptible (even so, the edge map remains almost intact!).
> >
> > We have also tested the proposed defense against CW and adaptive attacks shown in appendices J and L, respectively. Interestingly, shape defense withstands both types of attacks.
> >
> > Overall, we have conducted a comprehensive study to investigate the utility of shape for building robust models. The proposed defense is orthogonal to ideas in adversarial defense and is based on a very solid previous finding that shows CNNs are texture biased. This is in contrast to human vision and signifies a shortcoming with the convolution operation. Our work is an important step in mitigating texture bias and using shape in building robust models.

---

> ### Author Response · Authors · 2020-11-23
> **Updated manuscript in response to comments**
>
> We have updated the manuscript by performing an array of additional attacks focusing on adaptive attacks. Please see the updated pdf. More details are given in response to other reviewers raising similar concerns, in particular robustness against adaptive attacks.

---

### Official Review · AnonReviewer4 · 2020-11-02
**important topic, some concerns about the proposed methods**

**Rating:** 5
**Confidence:** 4

**Review:**

This paper studies how to incorporate shape (particularly depth map) into CNN for more robust models. The study focuses on image classification. Specifically, this paper proposes two depth-map-based defense: 1) Edge-guided Adversarial Training (EAT), which use depth map as an additional input 2) GAN-based Shape Defense (GSD), which learns a generator from depth map to reconstructed images, which is then used as net input. Experiments on 10 datasets shows the effectiveness of the proposed two defenses against white-box attacks including FGSM and PGD40. To further demonstrate the effectiveness, the authors also conduct some other experiments: 1) the proposed EAT goes well with two fast AT algorithms; 2) the proposed algorithm can also be used to defend backdoor attack; 3) edge makes CNN more robust to common image corruptions.

I think the topic that tries to explore and understand the connection between shape and CNN is very important and somewhat under-studied. This paper provides some interesting empirical results and insights to the community.

1. The assumption of this paper is that edge map does not change much under adversarial attacks. I think this relies on two things: A) we use tradition non-deep-net based edge detector like canny edge; B) the adversarial perturbation is pixel-based perturbation. I am curious to see how the proposed algorithm work with B), but for now, I will focus on A) below.

For A), for harder and more realistic datasets, canny may not work well and we may resort to deep-net-based edge detector (like HED). I think the author also mentions this for cifar10 and tinyimagenet, which is only 32x32 and 64x64. This problem likely becomes more severe when we deal with larger real images. But, if we use deep-net-based edge detector, we now break the assumption that edge map does not change much under adversarial attacks. Since it is deep net and so it is fragile. So I am not sure how the proposed method work with deep-net-based edge detectors, on a somewhat more realistic image datasets.

2. For GSD, as mentioned by the authors, it is similar to the two GAN/VAE-based baselines. I am curious to see how it compares with them?

3. Also for GSD, considering scaling up to more realistic images with more visual patterns, it would be super hard to learn a mapping from pure depth to rgb, since it is ill-posed and under-determined. The two baseline methods do not have this under-determination because their input is rgb images. Also, the learned mapper would be very correlated with and overfitting to the training data, which hinders generalization.

4. How does CW attack perform against the proposed defense?

5. A minor thing about reference. For the fast AT algorithms, to my best knowledge, I think there is a third one "Bilateral Adversarial Training: Towards Fast Training of More Robust Models Against Adversarial Attacks" published in ICCV 2019.

---

> ### Author Response · Authors · 2020-11-21
> **Concern regarding using a deep edge detector and more**
>
>
> Thanks for your review and finding our work interesting and important.
>
> 1. Regarding B) a pixel-based perturbation (or modifying a few pixels for that matter), has to be sharp enough to change the edge map and consequently the classifier built on the edge map. Such a sharp modification to the pixel value may make the perturbation perceptible (assuming that edge detector matches with human perception of edges and shape).
>
> Regarding A) Great point! Indeed, using a deep edge detector which tend to capture more salient and perceptually appealing edges, may improve the results, and as you mentioned breaks the system since the entire pipeline is differentiable now. The key here is that we use the Canny method which generates binary edge maps. If a deep edge detector also generates a binary edge map and is used inside the  image —> edge detector —> classifier pipeline, the attacker then can only back propagate to the image if he uses a differentiable function (eg sigmoid) at the end of the edge detector. Crafting an adversarial image for the new system, will likely perturb the pixels slightly (it won’t change the main edges to keep the perturbation imperceptible). Submitting the adversarial image to the original pipeline that uses the binarized version of the deep edge detector then will not fool the classifier.
>
> We have conducted an experiment that used HED edge detector to build adversarial images. Adversarial images crafted for this pipeline do not fool the original classifier build on Canny edge maps. Please see Appendix L.
>
>
>
> 2. Compare with DefenseGAN paper, our results are slightly lower than theirs but it is mainly because the classifiers that we use have lower standard accuracy (eg., over MNIST the standard accuracy of their model is 99.0+ whereas ours is around 96.0+, over MNIST their classifier has around 92% accuracy where as we get around 88%). This explains why their robust model perform a bit better than us. They do not however report results on CIFAR and other natural scenes datasets, whereas we do. It might not be easy to find the correct z corresponding to the clean image closest to the perturbed image using DefenseGAN over natural scenes. In our method, we explicitly use shape information which makes it easier to explain and aligns with other works highlighting the importance of shape in object recognition. Finally, as is also mentioned in the DefenseGAN paper, “The success of Defense-GAN relies on the expressiveness and generative power of the GAN. However, training GANs is still a challenging task and an active area of research, and if the GAN is not properly trained and tuned, the performance of Defense-GAN will suffer on both original and adversarial examples”.
>
>
> 3. Do you mean ‘edge map’ by ‘depth’?
>
> The quality of generated data depends of GAN performance. Even without conditioning on edge map some recent GANs (eg., BigGAN) are able to generate high quality cluttered natural scenes. Conditioning on an edge map (even imperfect) makes the job easier. As we have discussed using better GANs will improve the presented results in Fig. 4. The two baselines use the original image but they are not robust! Also, regarding your comment “the learned mapper would be very correlated with and overfitting to the training data, which hinders generalization.”, why would be the case if generated images have high quality (for example over MNIST and FashionMNIST). Finally, the idea of purification GAN has of course limitations. Its accuracy will depend on the type of data. If a GAN works well on a certain dataset the GSD would be a good defense! So, future progress on GAN will result in better GSD. Low performance on GSD on some problems (again we have not optimized the method by exploring all GANs) does not mean this method is not worth further exploration.
>
>
>
> 4. We performed an additional experiment to test the robustness of the proposed method against CW. Results shown in Appx. J show that shape defense is robust to CW attack.
>
>
> 5. Thanks for mentioning this paper. We will cite it in the version of the paper.

---

> > ### Author Response · Authors · 2020-11-23
> > **Using a deep edge detector ...**
> >
> > Continuing from "Regarding A) Great point! .... will not fool the classifier.",
> > We actually trained a deep edge detector to approximate the Canny edge detector on Canny and used that to generate adversarial examples for the Canny edge classifier (img --> edge --> label). Results are shown in Fig. 33 in Appx. M. As can be seen the adaptive attack created this way, is not able to fully knock down the model. This is mostly due to the fact that Canny generates a binary map thus effectively removing the noise from the adversarial example (i.e., edge maps stays the same).

---

> > ### Comment · AnonReviewer4 · 2020-11-23
> > **still some concerns**
> >
> > I appreciate the additional results and explanations from the authors! My concern is mainly about how the edge-map-based method scales to more real, higher resolution images.
> >
> > 1. I have some doubt on the experiment setup. Seems like you generated the adversarial example using HED but tested it with the pipeline using canny. It is only testing the weak transferability of the two detectors, which is expected since one is rule-based and one is learning-based. My original question is that: for real images, at least 300x400 like Imagenet, Pascal, COCO, canny may not be good and it may hurt the clean performance when concatenated with image channels as input (not sure of this but possible if the edge map is low-quality, then the help would be slim). So we have to use some advanced edge detector like learning-based HED. Then the system is no different from the typical differentiable deepnet and thus fragile.
> >
> > 2&3: Yes, I mean edge map, not depth map.
> >
> > " why would be the case if generated images have high quality (for example over MNIST and FashionMNIST)." It would be much more convincing using 300x400 natural images instead of MNIST.

---

> > > ### Author Response · Authors · 2020-11-23
> > > **Your concern regarding higher resolution images and edge maps**
> > >
> > > 1. Yes, you are right. We have conducted another experiment to approximate the Canny edge detector. Results are shown in Appendix M.
> > >
> > > Regarding resolution: This is exactly why we have included 10 datasets in the paper to make sure using edge map indeed helps.
> > > Some of the used datasets are high resolution (e.g., DogBreeds). It is not just about the resolution and image clutter matters as well. Point is when adding edge map does not help (\ie and does not hinder clean performance much; as we show is the case), it leads to higher robustness.
> > >
> > > CIFAR10 images are 32 x 32 and edge detection is really challenging on them, yet as you can see in Table 2, Img + Edge performs on par with the Img model and the full model is much more robust than the Img model. Same story happens over the tiny imagnet dataset with resolution of 64 x 64 (Table 5). So, even low quality edge maps still help (i.e., it has useful information that a classifier can latch onto).
> > >
> > > So, from our experiments, we have learned that augmenting edge map is most effective when scenes are not cluttered (eg. dog vs. cat; dog breeds, GTSRB, ...). And this is motivation behind the experiments in section 6 and 7, which says limiting the image area to the most important region for classification leads to robustness not only against classic evasion attacks but also backdoor attacks. Limiting the focus of attention to a single object will also lead to a more robust and useful edge map.
> > >
> > >
> > > Also, please notice that majority of the deep learning research is still using images below 300/400 pixels (with exceptions like segmentation tasks) to harness the computation.
> > >
> > > All in all, answering when and where using the Canny edge detector helps depends on both resolution and image clutter. And when Canny is not an option, then one might use other techniques and even robust deep edge detectors. What the community is certain about is that shape is very important to gain robustness and somehow should be taken into account. Here, we explored some possibilities which are very promising. As alternatives to edge augmentation, our works encourages other ways to emphasize edges (such as normalization layers as is discussed in the discussion section). Last but not least, we do not rule out using deep edge detectors in our pipeline even though they are differentiable. For example, one might decide to threshold the deep edge map before sending it to the classifier. The adversary still has access to the gradients to craft an adversarial image but once it is passed through the edge detector it is thresholded at inference time (just an idea!).
> > >
> > >
> > >
> > > 2&3: People are able to generate high quality faces from the face sketches using GANs over the CelebA dataset which contains high resolution images. We suspect using Canny edges will also lead to high quality conditional image generation on CelebA (and other datasets). Actually, our results using GSD on CIFAR 10 are very promising in the sense that even image generation from 32  x 32 edge maps on CIFAR10 still generates image that are useful. Working with high resolution images around 400 pixels as you mentioned really takes a lot of effort (to train a good GAN) and goes beyond the scope of our work. In experiments in section 4.2.2 we intended to show the possibility of coming up with a robust model using a conditional GAN. Also, one maybe able to use even a deep edge detector (with binary thresholding) to train a better cGAN and hence a better classifier with both high clean accuracy and robustness. Instead of spending a lot of time perfecting the cGAN, we have a proposed a universal solution based on shape and edge information that can be used for several purposes such as defending evasion attacks, hardening other defenses such as adversarial training, defensing backdoor attacks, improving robustness against common image corruptions, etc. We have shown these over a wide variety of datasets and experimental conditions with an exhaustive analysis.
> > >
> > > Thanks,

---

### Official Review · AnonReviewer1 · 2020-11-10
**Adversarial robustness using shape details in the form of edge maps**

**Rating:** 6
**Confidence:** 4

**Review:**

Summary: This paper aims to improve adversarial robustness considering the information about the object shape details with the means of edge maps. Two different strategies are proposed to increase model robustness using the edge maps: i) conduct the adversarial training on the input images, which are concatenated with its corresponding edge map as an additional input channel to the image. Here, the edge maps are recomputed and concatenated to the adversarial inputs after their generation during adversarial training. ii) Utilize a conditional GAN to generate the images from clean data distribution that is conditioned on the edge maps and later the classifier performance is evaluated on the generated images. Here, the authors claim that these two strategies improves the classifier robustness after conducting experiments across 10 different datasets. They also studied the effectiveness of their strategy when combined with background subtraction, the defense against poisoning attacks and robustness against natural image corruptions.

Strengths:
+ Motivation is clear.
+ The proposed strategies are interesting, explained clearly, easy to follow and are different from existing works.
+ Rigorous experiments across 10 different datasets are carried out to demonstrate the effectiveness of the proposed strategies.
+ Results demonstrate that the model robustness can be significantly improved using proposed strategies utilizing the edge maps. The improvement not only limited to adversarial robustness but also extends to backdoor attacks and natural image corruptions.

Weaknesses:
-	The major concern lies in the evaluation of the proposed strategies. Here, the authors considers that their method purify the input image before passing it to the model and an adaptive attack against their edge map based defense strategies will likely results in structural damage to the edge map. However, it is crucial to evaluate the proposed defense against an adversarial attack which craft the adversarial examples to produce minimal structural alterations to the edge map but mislead the model predictions. An adversary could potentially optimize the perturbation in such manner and may remain successful in attacking the model.
-	Results on CIFAR-10 and Icons-50 of GAN bases shape defense depicted in Figure 4 do not provide solid evidence on the model robustness. Here, the performance on clean inputs degraded significantly and the improvement seen in perturbed samples might be the result of the trade-off between model robustness and generalization as noted in the literature.
-	The claims on robustness against natural image corruptions using the edge maps seems to be valid only on GTSRB dataset and do not hold true for TinyImageNet and Icons-50 as seen in Figure 6. The robustness of the model with edge maps is similar or on par with the model without edge maps on these two datasets. These results suggest that additional usage of edge maps do not improve model robustness and the improved performance seen on GTSRB could be attributed to the simple nature of the objects in the dataset.

Final thoughts:
The proposed method is clearly motivated. Although the performance gains on adversarial robustness is significant, there are critical points yet to be addressed. Therefore, I marginally accept this paper.

------------------------------------------------------------------------------------------------------------
Post rebuttal:
The authors have devised an adaptive attack to craft the adversarial examples against edge maps and shown that the proposed technique is still remain robust. However, the essence of robustness in this work lies in the BINARIZATION of the input (i.e., binarized edge maps) which is shown in the previous work [1] and need not necessarily attribute to the shape information obtained through edge maps. I recently came across state-of-the-art deep edge detector [2] that produces non-binarized edge maps, which could be interesting for authors to validate their approach using such non-binary inputs. Hence, I maintain my initial rating and marginally accept this paper.


[1] ON THE SENSITIVITY OF ADVERSARIAL ROBUSTNESS TO INPUT DATA DISTRIBUTIONS, ICLR 2019

[2] Richer Convolutional Features for Edge Detection, CVPR 2017

---

> ### Author Response · Authors · 2020-11-12
> **evaluation against adaptive attacks and results on natural image corruptions**
>
> Thanks for finding our work interesting and for your review.
>
> Regarding evaluation of the proposed strategies: Indeed, we believe that any perceptible adaptive attack against our edge map based defense strategies will likely result in perceptible structural damage to the edge map (and hence easily detectable by the eye). Designing attacks to add minimal (and imperceptible) structural alterations to the edge map is non-trivial and perhaps not easy to come up with. However, one possible approach that we can think of is to use a deep edge detector in the pipeline and attack the entire pipeline with an adversarial attack. We will add the results here soon.
>
>
> GSD results on CIFAR-10 and Icons-50: We did not really try to optimize the GSD method to achieve the best image generation here. It is more like a proof of concept on natural scenes. We used the very basic image translation method (pix2pix) without perceptual loss. Performance of the GSD approach depends on the dataset as well. For example, constructing faces from face edge map is much easier since faces are usually registered at eyes. For natural scenes, it is harder but some recent methods are very promising (BIG GAN and StyleGAN). Nevertheless, as we show even without perfect image generation, models are more robust compared to the classifier trained on the original images. The trade-off between accuracy and robustness is mainly because of the imperfect conditional image generation. In other words, generating better images is going to improve both accuracy and robustness (no trade-off).
>
>
> Robustness against natural image corruptions: Perhaps not very pronounced, but across the three datasets there is at least one case where incorporating edges outperforms the original image-only classifier. Notice that we are using the Canny edge detector, and this is why results are better over GTSRB. With better edge detectors (e.g., deep ones), it is expected to see better results. Overall,
> our results are in alignment with Geirhos et al. (2018) where they showed ResNet-50 trained on the Stylized-ImageNet dataset performs better than the vanilla ResNet-50 on both clean and distorted images.
>
>
> As you mentioned, our approach is orthogonal to the existing works in adversarial defense and introduces a new idea which is using shape information. Shape and edge are invariant to adversarial perturbations (and also common image corruptions) as shown in Fig. 1 and are important features based on which humans classify objects and process scenes. Conversely, CNNs rely more on texture. We believe this is perhaps the main reason why CNNs are so brittle. Moving forward in object recognition, it seems inevitable to use shape information to achieve robustness.

---

> > ### Author Response · Authors · 2020-11-18
> > **Update on adaptive attack**
> >
> > According to your suggestion we now use a deep edge detector (holistically-nested edge detector; HED) to devise an adaptive attack. Please see Appendix L in the paper. In short, we used the HED edge detector and built a classifier on it. We then fed the generated adversarial images to the original model (based on the Canny edge detector). Even without adversarial training, the original model is robust to such adversarial examples (some are given in Fig. 30 in Appx L). We expect better results using adversarial training. The results shown in row 2 in Table 29 are about the same as when using the original model shown in Table 17 which shows that attack is effective around 60% of the time on the used dataset (so adaptive attack has not been successful). We believe the reason why the proposed defense works against such an adaptive attack is because it generates a binary edge map which retains the most important (and perhaps minimal information) from the image to classify the input. Any adaptive attacks has to change the image such that its Canny edge map is significantly changed (notice again that the edge map is binary not real-valued).

---

> > > ### Comment · AnonReviewer1 · 2020-11-18
> > > **It is related to a previous work**
> > >
> > > Thanks for conducting an experiment with an adaptive attack. As you mentioned that the generation of binary edge map is the reason for the successful defense, a previous work [1] also studied this input sensitivity and shown that such technique improves robustness. Then how the defense proposed in the paper different than the previous work [1] ?
> > >
> > > [1] ON THE SENSITIVITY OF ADVERSARIAL ROBUSTNESS TO INPUT DATA DISTRIBUTIONS, ICLR 2019

---

> > > > ### Author Response · Authors · 2020-11-18
> > > > **Difference with mentioned work**
> > > >
> > > > Thanks for pointing out this paper. It is actually quite interesting and relevant to the spirit of our study.
> > > > The main finding in Ding et al.'s work is that adversarial robustness is sensitive to semantically lossless shifts in input data distribution. More specifically, input saturation leads to both high standard and robust accuracies. For example, for binarized MNIST with adversarial training, the clean accuracy and the robust accuracy are almost the same (both high), which in turn means that getting high robust accuracy on binarized MNIST does not conflict with achieving high clean accuracy. Similar observations have been made with input quantization.
> > > >
> > > > Input binarization (or saturation techniques) effectively reduce the dynamic range of the pixels and also lower the amount of texture. In order to perturb such inputs, then higher perturbation is needed. For example, reducing the range from {0,1,2,..,255} to {0,15,31,...255} or {0,255} in extreme case imposes a perturbation to modify pixels more drastically. For example, in the binary case, a pixel has to be toggled which has a sever effect on the image (and Lp distance). This hints maybe quantization and binarization emphasize more on the object structure and shape than texture. What we show here is one step further. A much smaller number of pixels are enough to achieve robustness and those are edge pixels. Edge map is another semantically lossless shift that can achieve both high accuracy and robustness. That being said, our work actually aligns with the mentioned paper. Here, however, we propose algorithms to operationalize the idea of using shape and edge as a defense.
> > > >
> > > > The simplest core idea would be performing classification only on the edge map (and hence discarding problematic texture). For this, a reliable and robust edge detection is needed which is not available to date. In human vision, an unknown robust normalization processing might be responsible. We propose to use edge map as an additional channel and recompute it at inference time. Edge map this way acts as a signature which does not change much (with perturbation)
> > > > and is carried along with the image. We showed that our method is robust to adaptive attacks (ie a differentiable edge detector) which is important. In sum, we will certainly discuss this paper in our work. It is interesting to see machine learning approaches to adversarial robustness converge to ideas from human visual processing and hopefully this will lead to robust algorithms. We envision shape to be the core element of such algorithms.

---

### Author Response · Authors · 2020-11-18
**To all reviewers (and please see the revised paper):**

A major concern which was raised was robustness against adaptive attacks. We have addressed this concern in the new submission.
Although it is not possible to explore all possible attacks against a defense, here we use a deep learning edge detector to approximate the non-differentiable Canny detector and then use it to construct adversarial examples. We find that the proposed defense is still robust to such attacks. Please see the paper for illustrations and discussions pertaining why this is the case.

Please also notice that diverging from the main stream in adversarial defense, here we propose a defense which is based on shape information (and hence tackles texture bias). As we discuss in the discussion section, there might be also other ways to mitigate texture bias (\eg using normalization layers). Our work provides a foundation to build upon and to utilize shape information in building future defenses. Our proposed defense agrees with human vision (which is also more biased towards shape) and offers a comprehensive solution to defend against a variety of attacks including backdoor attack (together with background subtraction).

Shape defense is tested over 10 datasets and parameters and efficiency are studied exhaustively.

Thank you.

---

### Decision · Program_Chairs · 2021-01-07
**Final Decision**

**Decision:**

Reject

**Comment:**

The paper proposed a method for adversarial robustness by considering information from the edge map of the images. Two reviewers point out the similarities of the paper with previous work ([1]) and it is unclear whether the benefits come from binarization of the input or from shape information. As such, the paper is not suggested for publication at this time